# Sloppiness: Fundamental study, new formalism and its application in model assessment

Prem Jagadeesan[1,3,4], Karthik Raman[2,3,4]*, Arun K. Tangirala[1,3,4]*

**1** Department of Chemical Engineering, Indian Institute of Technology (IIT) Madras, Chennai, India,
**2** Department of Biotechnology, Bhupat and Jyoti Mehta School of Biosciences, IIT Madras, Chennai, India,
**3** Robert Bosch Centre for Data Science and Artificial Intelligence (RBCDSAI), IIT Madras, Chennai, India,
**4** Centre for Integrative Biology and Systems mEdicine (IBSE), IIT Madras, Chennai, India

* kraman@iitm.ac.in (KR); arunkt@iitm.ac.in (AKT)

**Data Availability Statement:** All relevant data are within the paper and its Supporting information files.

## Abstract

Computational modelling of biological processes poses multiple challenges in each stage of the modelling exercise. Some significant challenges include identifiability, precisely estimating parameters from limited data, informative experiments and anisotropic sensitivity in the parameter space. One of these challenges' crucial but inconspicuous sources is the possible presence of large regions in the parameter space over which model predictions are nearly identical. This property, known as sloppiness, has been reasonably well-addressed in the past decade, studying its possible impacts and remedies. However, certain critical unanswered questions concerning sloppiness, particularly related to its quantification and practical implications in various stages of system identification, still prevail. In this work, we systematically examine sloppiness at a fundamental level and formalise two new theoretical definitions of sloppiness. Using the proposed definitions, we establish a mathematical relationship between the parameter estimates' precision and sloppiness in linear predictors. Further, we develop a novel computational method and a visual tool to assess the goodness of a model around a point in parameter space by identifying local structural identifiability and sloppiness and finding the most sensitive and least sensitive parameters for non-infinitesimal perturbations. We demonstrate the working of our method in benchmark systems biology models of various complexities. The pharmacokinetic HIV infection model analysis identified a new set of biologically relevant parameters that can be used to control the free virus in an active HIV infection.

## Introduction

The dynamics of biological systems are generally considered complex owing to their non-linearity, interconnections and multi-scale nature. Complex dynamical systems are often modelled as non-linear ordinary differential equations (ODEs) with many states and parameters. Dynamical models of biological systems are central to many applications. Recent advances in

**Funding:** PJ acknowledges funding from the Ministry of Education, Government of India. The funders had no role in study design, data collection and analysis, decision to publish, or preparation of the manuscript.

**Competing interests:** The Authors declare no Competing Financial or Non-Financial Interests.

whole-cell modelling [1] and quantitative systems pharmacology (QSP) [2, 3] underscore the rising importance of dynamical modelling for complex biological systems. Identifying such a complex dynamical system from data poses multiple challenges in each stage of the identification exercise [4]. Identifiability, precise parameter estimation and an-isotropic sensitivity are a few crucial challenges in the computational modelling of complex dynamical systems. One of the crucial and often overlooked sources of these challenges is the possible presence of large regions in the parameter space over which the model predictions are nearly identical. This phenomenon is known as *sloppiness* [5]. Sloppiness is closely related to another property known as *identifiability*, where multiple parameter sets result in identical predictions [6].

Identifiability is a well-established concept in the domain of system identification. Loss of identifiability can result from either the model structure (structural identifiability) or data [7, 8]. The loss of structural unidentifiability implies multiple solutions to the estimation problem. Thus, checking for structural identifiability before parameter estimation is imperative. There are several analytical methods available for assessing the structural identifiability of a model structure [6, 9–13]. A differential geometric approach using observability condition is proposed in [14]. However, most of the analytical methods are not scalable to large models. A numerical method is proposed in [15] to assess local structural unidentifiabilities. A global method has been proposed to detect identifiability, active subspaces and parameter sensitivity exploiting the underlying relationship with sensitivity Fisher Information Matrix (sFIM) is proposed in [16].

While identifiability is a 'binary' situation, sloppiness lies somewhere between identifiability and loss of identifiability; moreover, the closer it is to the loss of identifiability, the more problematic it is. The presence of sloppiness often results in huge uncertainties in parameter estimates [5]. To illustrate sloppiness consider the simple bi-exponential model in (1), with a true parameter vector, $\theta = \begin{bmatrix} 1 & 10 \end{bmatrix}^T$. The model output $y(t)$ is computed for 5 seconds by fixing one parameter and varying the others at a time.

$$y(t) = e^{-\theta_1 t} + e^{-\theta_2 t} \tag{1}$$

From Fig 1a, it is evident that for a range of $\theta_2$ values in parameter space, the model predictions are nearly identical, and hence, $\theta_2$ direction is considered sloppy. However, in Fig 1b, the change in parameter $\theta_1$ results in clearly distinctive model outputs. Here, $\theta_1$ is a *stiff* direction. From Fig 1c, it is seen that there are certain directions over which the model predictions are nearly identical yet significantly vary in other directions. However, in many cases, instead of individual parameters being sloppy and stiff, there will be directions in the parameter space that are sloppy and stiff. The nearly identical model outputs for a significant range of parameter sets might reflect as large standard errors in a subset of parameter estimates. These large standard errors in the parameter estimates, often referred to as practical/numerical identifiability [17] is one of the crucial challenges while modelling sloppy models [5, 15, 18].

Practical identifiability is formally defined as the ability to estimate parameters of a structurally identifiable model with acceptable precision given a data set [17, 19]. Practical identifiability is function of both data (inputs and measurement errors) [20] and model structure. Practical identifiability/precision of the parameter estimates and their relationship with input and cost function has been extensively studied in the domain of non-linear system identification [7, 21]. Practical identifiability is assessed by the width of the confidence interval of the parameter estimate [17]. However, there are other numerical methods proposed to assess practical identifiability [11, 22, 23]. The relationship between structural identifiability and sloppiness is well-established in [15]. A sloppy model is always structurally identifiable. Whereas in the case of practical identifiability, though sloppiness is closely related to practical identifiability, the exact

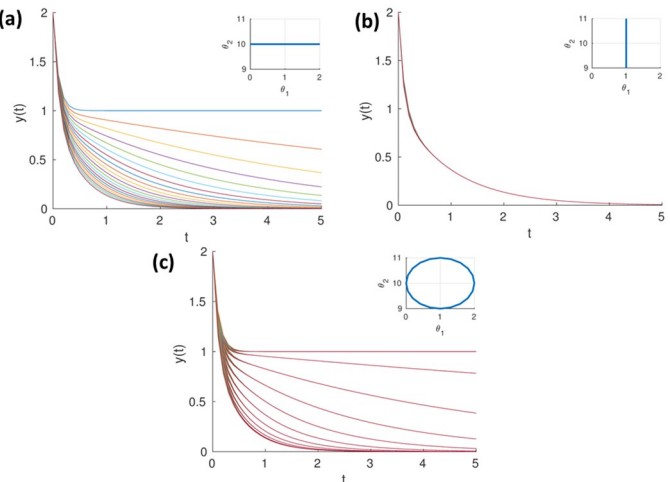

**Fig 1.** (a) shows the model output while varying $\theta_2$ from 9 to 11 and fixing $\theta_1 = 1$. The output of the model does not significantly vary and is qualitatively indistinguishable. (b) shows the model output while varying $\theta_1$ from 0 to -2 and fixing $\theta_2 = 10$. The output significantly varies while changing $\theta_1$ in the same range. (c) shows the outputs for a range of parameters varied over the circumference of a circle with a fixed radius, $r = 1$ and $\theta_1 = 1$, $\theta_2 = 10$ as center. Visually evident chunks of outputs that are nearly identical. This implies certain directions in the parameter space over which model outputs are nearly identical.

relationship is still unclear [15]. This has been one of the sources of ambiguity in understanding the impact of sloppiness analysis as a part of any modelling exercise. The relationship between sloppiness, structural, and practical identifiability is depicted in the Fig 2. Sloppy models intersect with practically unidentifiable models. However, how much sloppiness affects practical identifiability is the question that needs to be answered.

## Impact of sloppiness in optimization, experiment design and model validation

Along with structural and practical identifiability, impact of sloppiness on various facets of modelling has been extensively studied in the past decade [5, 15, 24–28]. The impact of sloppiness on designing optimal experiments, optimization algorithms, and uncertainty quantification has been addressed widely in a variety of studies, as we discuss hereon. It has been shown that sloppy models have nearly flat cost surfaces in the vicinity of the optimal parameter set

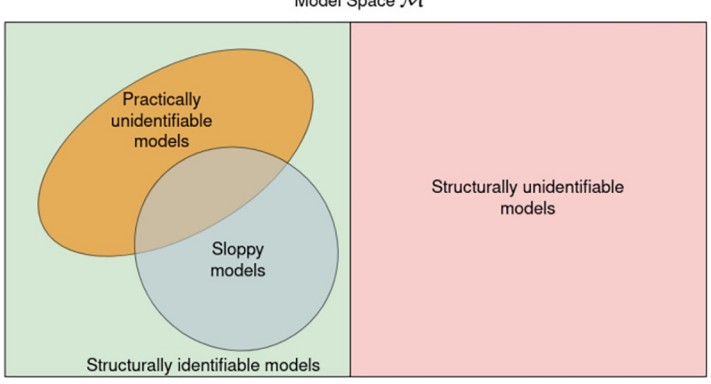

**Fig 2. The relationship between structural identifiability, practical identifiability, and sloppiness.**

[24]. In such cases, it is observed that non-linear optimization algorithms may get trapped in sloppy regions. A differential geometric approach has been employed to improve the convergence of the Levenberg-Marquardt algorithm. In the case of validating models, it is observed that the uncertainty estimates are unreliable in the case of sloppy models [25]. Henry *et al.* show that maximizing mutual information between uninformative prior and posterior distribution in Bayesian estimation effectively selects simple models [29]. They have suggested modified Markov Chain Monte Carlo (MCMC) simulations to circumvent this issue. An attempt to design optimal experiments for a precise estimate of parameters in sloppy models may result in compromising prediction accuracy [18].

Numerous attempts have been made to resolve the issues arising from sloppy models, but very few works have attempted to find the root cause of sloppiness [17, 26, 30]. Though the source of sloppiness is attributed to both model and data [5], in most cases, the source of sloppiness is attributed to the model structure [5, 15, 18, 25, 28, 31, 32]. A computational study has been carried out to study the relationship of sloppiness measure with structural identifiability, practical identifiability, and experimental design [17]. The result suggests that the relationship of sloppiness measure with practical identifiability is inconclusive. Tönsing et al [30] have shown that the root causes of sloppiness include both the model and experimental conditions. They suggest that by changing the experimental condition, it is possible to cure sloppiness. Apgar et al [31] have shown it is possible to estimate parameters precisely for sloppy models, by careful design of experiments. Sloppiness has also been reported to have connections with biological phenomena such as robustness and evolvability [11]. Ben Matcha *et al.* suggest in [33] that the sloppiness is essentially an effect of the multi-scale nature of the underlying process. Further, they show that specific models are not sloppy at microscopic scales, and sloppiness emerges only when fit to collective behaviour. Transtrum *et al.* and Katherine *et al.* suggest that the existence of sloppiness (dependence of model parameters to only a few macroscopic parameter directions) is the origin of simplicity in science and responsible for the emergence of comprehensible macroscopic theories from highly complex microscopic processes and [34, 35]. Evangelou *et al.* have developed a method to identify important and unimportant parameter combinations using manifold learning techniques (Dmaps). Further, they use a special type of auto-encoder known as a Y-shaped conformal auto-encoder to disentangle the unimportant parameter combinations. Finally, the identified effective parameters are mapped back to physical parameters [36].

The utility of sloppiness analysis in modelling has been questioned in [4, 17]. These studies argue through computational study that the presence of sloppiness does not guarantee either structural or practical unidentifiability. Based on the results, they conclude that using sloppiness analysis to assess the identifiability of parameters can be misleading and suggest identifiability analysis as a better tool. The relationship of sloppiness with Fisher information-based experiment design criteria has also been inconclusive. Though the study is convincing, the effect of sloppiness on errors in the parameter estimate is inevitable. The relationship of sloppiness with practical identifiability is widely accepted but not formally established [15, 17].

In all of the above studies, the exact role of sloppiness on various aspects of modelling, such as precision of the parameter estimates, prediction uncertainty and experiment design, is mainly inconclusive because of the lack of formal mathematical relationship between the measure of sloppiness and other properties. A formal mathematical definition of *sloppiness* is the missing piece in the puzzle. In order to mathematically formalize sloppiness, we first give two motivating examples to point out that with the existing measure of sloppiness, in the case of non-linear predictors, it is not possible to attribute sloppiness to model structure alone decisively. In order to circumvent the ambiguity, in this work, first, we formulate two new theoretical definitions of sloppiness: (i) sloppiness and (ii) conditional sloppiness. The conditional

sloppiness is conditioned on the experiment space. The proposed definitions of sloppiness for autonomous ODE systems are defined in an augmented space of parameters and initial conditions. This implies that both model and experimental conditions together are responsible for sloppiness. Secondly, we establish a mathematical relationship between practical identifiability and sloppiness in the case of linear predictors.

Further, we propose a scalable computational method based on the proposed new definitions of sloppiness to assess the goodness of model structure in viable parameter space. The proposed method facilitates detecting conditional sloppiness, insensitive parameters, and local structural unidentifiability. The proposed method can be used to assess the goodness of the model structure in the viable space before estimation. The benefit of analyzing the model in the viable space is two-fold—once we know that the viable space of the model is in a sloppy region, the probability of the estimated model landing in the sloppy region is high. In such cases, appropriate experiments can be designed to circumvent the effects. On the other hand, if the viable space of the model (valid values of parameters and experimental conditions) is not in a sloppy region. If the estimated model is sloppy, we can fine-tune the estimation algorithm and other controllable experimental conditions. The proposed tool can also detect multi-scale sloppiness described in [15].

## Motivating examples

This section provides two motivating examples that illustrate the challenges in the current measure of sloppiness in answering the questions of interest in this work. Following are the two challenges in the existing measure of sloppiness (i) Assessing the role of model structure in sloppiness with the existing sloppiness measure and (ii) Detecting practical identifiability with the existing sloppiness measure. We consider a non-sloppy model and show that it can be turned sloppy by changing experimental conditions, and a model that is regarded as sloppy can be made non-sloppy again by changing the experimental condition.

**Example 1: The source of sloppiness in linear predictors is data.** Consider the linear predictor given in (2). Let $\mathbf{z}$ be a vector of $m$ observations. The parameters of the model are estimated using the method of least squares. The Hessian of the cost function is given in (4).

$$y = a_1 x_1 + a_2 x_2 + \ldots + a_n x_n \tag{2}$$

Now, let us define

$$C(a) = \sum_{i=1}^{m} (z_i - y)^2 = \sum_{i=1}^{m} (z_i - (a_1 x_{1_i} + a_2 x_{2_i} + \ldots + a_n x_{n_i}))^2 \tag{3}$$

$$\nabla^2(C(a)) = \begin{bmatrix} 2\sum_{i=1}^{m} x_{1_i}^2 & 2\sum_{i=1}^{m} x_{1_i} x_{2_i} & \cdots & 2\sum_{i=1}^{m} x_{1_i} x_{n_i} \\ 2\sum_{i=1}^{m} x_{2_i} x_{1_i} & 2\sum_{i=1}^{m} x_{2_i}^2 & \cdots & 2\sum_{i=1}^{m} x_{2_i} x_{n_i} \\ \vdots & \ddots & \ddots & \vdots \\ 2\sum_{i=1}^{m} x_{n_i} x_{1_i} & \cdots & \cdots & 2\sum_{i=1}^{m} x_{n_i}^2 \end{bmatrix} \tag{4}$$

For the purpose of discussion, we use an existing measure of sloppiness given in (5), the ratio

of smallest to largest eigenvalues of the Hessian of the cost function.

$$S = \frac{\lambda_{min}(x)}{\lambda_{max}(x)} \qquad (5)$$

Linear predictors are not generally observed to be sloppy [18]. However, in the above example, the ratio of eigenvalues is only a function of data and not the model parameters—hence, we can see that it is possible for the linear predictor to show sloppiness for some experimental conditions. Moreover, sloppiness in linear predictors given in the above example is purely an artefact of data/input and not due to the nature of the model structure itself.

**Example 2: Both data and model structure are responsible for sloppiness in nonlinear predictors.** Consider the state space model in (6). The Hessian of the least-squares cost function is given in (7).

$$\begin{bmatrix} \dot{x}_1 \\ \dot{x}_2 \end{bmatrix} = \begin{bmatrix} -\theta_1 & 0 \\ 0 & -\theta_2 \end{bmatrix} \begin{bmatrix} x_1 \\ x_2 \end{bmatrix} \qquad (6)$$

$$y(t) = x_1(0)e^{-\theta_1 t} + x_2(0)e^{-\theta_2 t}$$

$$\nabla^2(C(a)) = \begin{bmatrix} \int_0^t \theta_1^2 x_1(0)^2 e^{-2\theta_1 t} dt & \int_0^t \theta_2\theta_2 x_1(0)x_2(0)e^{-(\theta_2+\theta_2)t} dt \\ \int_0^t \theta_2\theta_2 x_1(0)x_2(0)e^{-(\theta_2+\theta_2)t} dt & \int_0^t \theta_2^2 x_2(0)^2 e^{-2\theta_2 t} dt \end{bmatrix} \qquad (7)$$

Sloppiness is computed in the grid $1 \leq \theta_1 \leq 100$ and $1 \leq \theta_1 \leq 100$. From Fig 3a, it is evident that sloppiness is a function of parameter space. In Fig 3b, sloppiness is computed over a grid of initial conditions $1 \leq x_1(0) \leq 100$ and $1 \leq x_2(0) \leq 100$ around the point $\theta_1 = 1$, $\theta_2 = 100$. The model is simulated for $t = 0$ to $t = 20$ seconds with a sampling interval of 0.5 seconds. The model is structurally identifiable for all the initial condition and parameters considered.

From Fig 3a, we can observe that the change in experimental condition can cure sloppiness, which indicates that sloppiness is a function of information contained in a data set. In our previous work [37], we have demonstrated using simulations that parameters contributing to

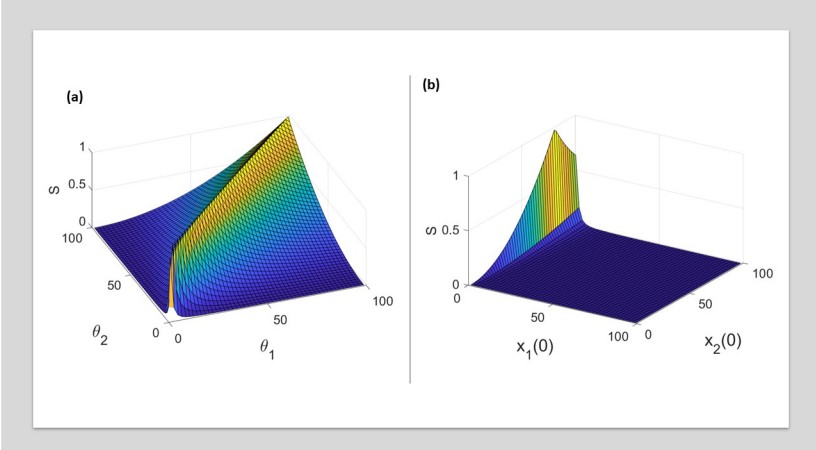

**Fig 3.** (a) Shows sloppiness in the parameter space. There are regions in the parameter space where the system is non-sloppy. (b) Shows sloppiness for various initial conditions. For a subset of initial conditions, the model becomes non-sloppy.

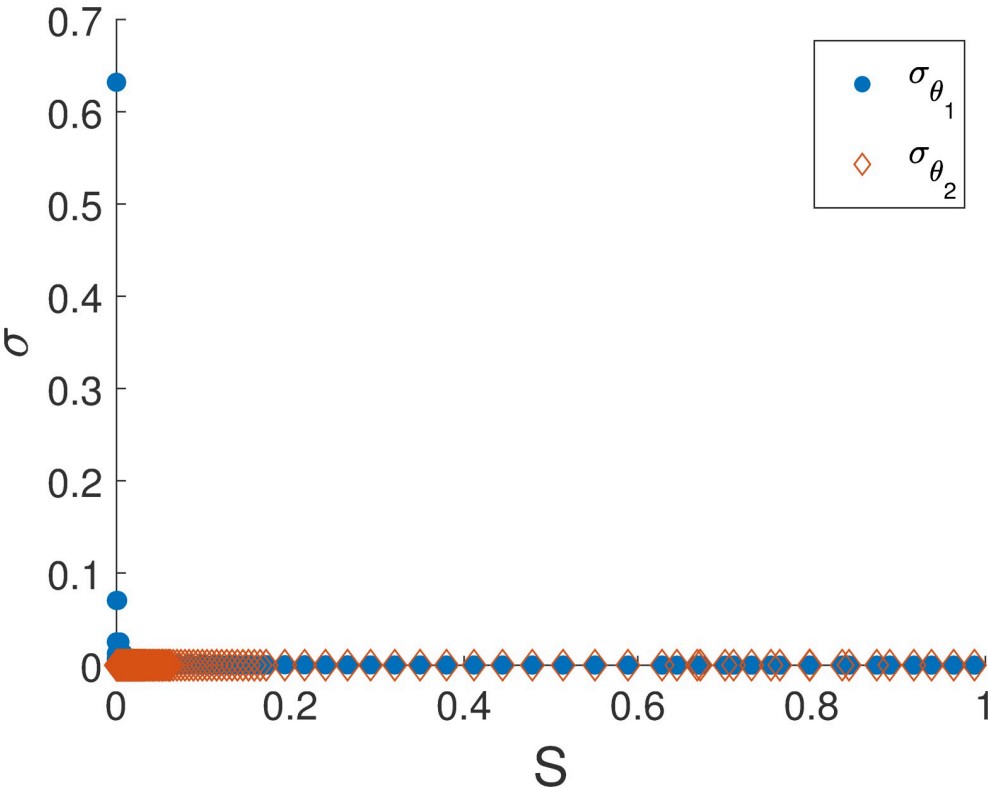

**Fig 4. Standard errors of the parameter estimates obtained from various initial conditions (Fig 3b) plotted against the corresponding sloppiness value.** It is seen that there is no specific relationship between sloppiness and standard errors.

sloppy directions have very low information gain in the Bayesian framework. For initial condition > 50 of $x_2$, the model becomes non-sloppy. This has been extensively studied in [11, 30, 31]. Tonsing *et al. (2014)* showed that the structure of the sensitivity matrix is the reason for sloppiness in ODEs and also concluded that both experiment and model structure are the root causes of sloppiness. Fig 4 shows that the relationship between sloppiness and standard errors of the parameters does not follow any definitive trend. Though it has been argued that sloppiness and practical identifiability are two distinct concepts and that they are incorrectly conflated [18], the effect of sloppiness on practical identifiability cannot be ignored [5].

Following are two important observations emanate from our study: (i) for a non-linear least-squares estimation problem, it is impossible to attribute sloppiness to the model structure alone. Sloppiness is a function of both parameter space and initial conditions, and hence, labelling a model to be sloppy with the current analysis method of sloppiness can be misleading [5, 18] (ii) sloppiness often results in loss of practical identifiability. However, the exact relationship is not revealed by the current measure of sloppiness.

The rest of the paper is organized as follows: Section 2 provides perspectives on sloppiness by revisiting the concept of sloppiness and rightly positioning it relative to identifiability. Section 3 presents key results, including two motivating examples to highlight the challenges in using the current measure of sloppiness. Further, it provides two new definitions of sloppiness and their relationship with practical identifiability in the case of linear predictors. Section 4 illustrates the proposed method to assess the sloppiness of non-linear predictors. The paper ends with concluding remarks in Section 5.

## Problem statement

The demonstrated challenges of the current measure of sloppiness results in the need to answer the following questions: (i) what is the source of sloppiness? (ii) what is the formal relationship of sloppiness with parameter uncertainty / practical identifiability? (iii) what is the relationship of sloppiness with inputs and estimation algorithm? In this work, we propose two new mathematical definitions of sloppiness to circumvent these challenges. Further, we develop an improved computational method to detect local structural identifiability and sloppiness along with sensitive and insensitive parameters in the viable region of the parameter space.

## Perspectives

### Nature of sloppiness

Sloppiness, at its core, quantifies the sensitivity of output with respect to changes in parameters or directions in the parameter space. When the gradient of the predictor qualitatively does not change as the parameter is varied significantly, then the system is sloppy. Sloppiness can be observed as pockets of regions in the parameter space; in such cases, the gradient of the predictor varies significantly in the parameter directions.

In the existing literature, a method of quantifying this insensitivity of predictions to changes in parameters is captured by (8).

$$S = \frac{\lambda_{min}(H)}{\lambda_{max}(H)} \tag{8}$$

where $H$ is the Hessian of the cost function, in general, for a non-linear parameter space, it is difficult to identify all such pockets of sloppy regions. Hence, sloppiness is determined locally around a parameter of interest by computing the sensitivity of the output to the change in the parameter. The sloppiness is characterized by equally spaced eigenvalues of the Hessian of the cost function in the log scale and is quantified by (8).

Even though there is no strict cut-off, a model is considered to be sloppy if $S \leq 10^{-6}$ [32]. The eigenvectors corresponding to maximum and minimum eigenvalues indicate the stiff and sloppy direction, respectively. It can be observed that the Hessian of the cost function is representative of the derivative of the output with respect to the parameters. The above measure of sloppiness is valid only if the estimation algorithm is least-squares; for a different estimation algorithm (for example, L1 norm), the Hessian of the cost function may not be representative of the sensitivity of the output with respect to the parameters.

### Sloppiness and identifiability

As a first step in answering the questions raised above, the notion of sloppiness needs to be rightly positioned relative to well-established concepts such as structural identifiability and practical identifiability. Additionally, when a model is sloppy, it is important to attribute the source of sloppiness to the appropriate factors.

Table 1 shows the list of model properties, their definitions, and the corresponding method of assessment. Even though qualitative definitions for practical identifiability and sloppiness are available, a formal mathematical framework for defining sloppiness is not available. In this work, we formalize sloppiness mathematically. Mathematical definitions are given in the results section.

All three properties of interest that are under discussion arise due to either one or more of the data, model, and estimation algorithm. The data itself is characterized by (i) signal-to-

**Table 1. List of model properties and their definitions.**

| # | Model Property | Definition | Assessment |
|---|---|---|---|
| 1 | SI | $\hat{y}(t, \theta_1) = \hat{y}(t, \theta_2) \Leftrightarrow \theta_1 = \theta_2$ | Direct application |
| 2 | PI | - | $\text{trace}(\Sigma_{\hat{\theta}})$ |
| 3 | Sloppiness | - | $H_{i,j} = \frac{\partial^2 C(\theta)}{\partial log\theta_j \partial log\theta_i}$ |

**Table 2. Factors influencing model properties.**

| # | Model Property | SNR | Sample size | Input | $\nabla(y(\theta))$ | $C(\theta)$ |
|---|---|---|---|---|---|---|
| 1 | SI | - | - | - | ✓ | - |
| 2 | PI | ✓ | ✓ | ✓ | ✓ | ✓ |
| 3 | Sloppiness | - | - | ✓ | ✓ | - |

noise ratio (SNR), (ii) sample size, and (iii) the input. While input and SNR determine the quality of data, the quantity of data is determined by sample size. The model's contribution towards these properties is characterized by predictor gradient, and finally, the estimation method is generally characterized by the cost function.

The relationship of sloppiness with practical identifiability is still an open problem [15]. From Table 2 it is clear that both the input and the gradient of the output with respect to parameters affect sloppiness and practical identifiability. This is the reason why most parameters in a sloppy model become practically unidentifiable when the gradient is large. In the following section, we provide this work's primary results: The mathematical formalism of sloppiness and its application to model assessment.

## Results

In answering the crucial questions about sloppiness, we formulate two new mathematical definitions of sloppiness. The proposed mathematical definitions of sloppiness are more accurate descriptions of sloppiness as they are formulated from the fundamental nature of sloppiness.

### A new mathematical definition of sloppiness

The above-perceived challenges of the present sloppiness analysis motivated us to formulate two new mathematical definitions of sloppiness. The new definitions of sloppiness are based on the following remarks.

**Remark 1**. *Sloppiness is assessed across an augmented space (parameters, initial conditions, and inputs) rather than parameter space alone. In the case of autonomous ODE models, the augmented space is*

$$\phi = \begin{bmatrix} \theta_1 & \theta_2 & \cdots & \theta_n & \vdots & x_1(0) & x_2(0) & \dots & x_m(0) \end{bmatrix}$$

**Remark 2**. *A significant change in the subset of augmented space (parameter space) results in a small change in prediction space.*

$\mathcal{D}_{\mathcal{M}}$—All possible parameter values the model $\mathcal{M}$ can take. $\mathcal{I}$—Identifiable region, $\mathcal{Z}_{\mathcal{M}}$— Set of all experiments for which the model structure $\mathcal{M}$ is identifiable. $\mathcal{S}$—of all parameters that belong to the sloppy region.

**Definition 1**. *A model $\mathcal{M}$ is $(\epsilon, \delta)$ sloppy with respect to an experiment space $\mathcal{Z}_\mathcal{M}$ at $\theta^* \in \mathcal{I} \subset \mathcal{D}_\mathcal{M}$, if*

$$||\theta^* - \theta||_2 > \delta \,\, \forall \theta \in \mathcal{S} \subset \mathcal{I} \tag{9}$$

$$||y(\theta^*, t) - y(\theta, t)||_2^2 < \epsilon \,\, \forall \mathcal{Z} \in \mathcal{Z}_\mathcal{M} \tag{10}$$

*for every $(\theta_1, \theta^*)$ satisfying (9) & (10). $\epsilon$ is arbitrarily small. $\delta \gg \epsilon$.*

**Definition 2**. *A model $\mathcal{M}$ is conditionally $(\epsilon, \delta)$ sloppy with respect to an experiment space $\mathcal{Z}_\mathcal{M}$ at $\theta^* \in \mathcal{I} \subset \mathcal{D}_\mathcal{M}$, if*

$$||\theta^* - \theta||_2 > \delta \,\, \forall \theta \in \mathcal{S} \subset \mathcal{I} \tag{11}$$

$$||y(\theta^*, t) - y(\theta, t)||_2^2 < \epsilon \,\, \forall u \in \mathcal{Z} \subset \mathcal{Z}_\mathcal{M} \tag{12}$$

*for every $(\theta_1, \theta^*)$ satisfying (11) & (12). $\epsilon$ is arbitrarily small. $\delta \gg \epsilon$.*

Eqs (9) & (10) say that a model is considered sloppy, if the $(\epsilon, \delta)$ condition holds true for all possible experimental conditions $\mathcal{Z}_\mathcal{M}$. Similarly, (11) & (12) convey that the model is conditionally sloppy if the $(\epsilon, \delta)$ condition holds true only for a subset $(\mathcal{Z})$ of all possible experimental conditions $(Z)$.

No longer the proposed sloppiness and conditional sloppiness are used in a generic sense; rather, they have to be used in conjunction with $\epsilon$ and $\delta$. By virtue of our definitions, we believe, by itself, the sloppiness is qualitative, and where ever quantitative sloppiness is to be discussed, the $(\epsilon, \delta)$ from should be used. For a large $\delta$ if the $\epsilon$ is negligibly small, then the system is considered to be $(\delta, \epsilon)$ sloppy. Note that the proposed measure of sloppiness is different from the multi-scale sloppiness proposed in [15]. Multi-scale sloppiness is the ratio of maximum to minimum prediction error for a non-infinitesimal perturbation from a reference parameter $(\theta^*)$, whereas, in the proposed definition, sloppiness is a function of both prediction error and parameter perturbation which is a more natural way to define and understand the notion sloppiness.

**Sloppiness analysis of a linear predictor.** Consider the linear predictor in (13)

$$y(x) = a_1 x_1 + a_2 x_2 + \cdots + a_n x_n \tag{13}$$

$$\theta = \begin{bmatrix} a_1 & a_2 & \cdots & a_n \end{bmatrix}^T \& \quad \mathbf{X} = \begin{bmatrix} \mathbf{x_1} & \mathbf{x_2} & \cdots & \mathbf{x_n} \end{bmatrix}$$

$$\mathbf{y} = \mathbf{X}\theta \tag{14}$$

The model is identifiable if $\mathbf{X}$ is full column rank. Consider another parameter vector $\theta_1$ and the corresponding prediction $\mathbf{y_1}$. The model is sloppy if $\mathbf{X}(\theta^* - \theta_1) = (\mathbf{y} - \mathbf{y_1})$ for $||\theta^* - \theta_1||_2 > \delta$ and $||\mathbf{y} - \mathbf{y_1}||_2 < \epsilon$.

Let $\theta^d = (\theta^* - \theta_1)$, $||\theta^d||_2 \gg \epsilon$.

Using matrix norm,

$$||\mathbf{X}\theta^d||_p \leq ||\mathbf{X}|| ||\theta^d||_p \,\, \forall p \in \mathcal{R}. \tag{15}$$

Consider the extreme case for $p = 2$,

$$||\mathbf{X}|| = \frac{\epsilon^2}{\delta} \tag{16}$$

For extremely small $\epsilon$ and extremely large $\delta$, the $||\mathbf{X}|| \approx 0$ and will become numerically

unstable and that will result in loss of identifiability. This is the reason why in most cases, linear least-square problems are observed to be non-sloppy. Once the system is found practically unidentifiable, then a careful design of the experiment will constrain the parameter uncertainty making the matrix norm of data significantly large. In such a case, the possibility of sloppiness is almost eliminated.

**Relationship between conditional sloppiness and practical identifiability.** For a structurally identifiable model structure $\mathcal{M}(\theta)$ and a data set $\mathbf{z}$, precision of parameter estimate $\theta_i \in \theta$ is a measure of practical identifiability. Practical identifiability is assessed for a data set $\mathbf{z}$ given a model structure $\mathcal{M}$. A parameter $\theta_i$ is said to be practically unidentifiable for a given $\mathbf{z}$, if

$$\sigma_{\theta_i} > \hat{\delta}_i$$

$$d(\mathbf{z}, y(\hat{\theta})) < \epsilon$$

where $\sigma_{\theta_i}$ is the standard error of the parameter estimate $\theta_i$ and $d(\mathbf{z}, y(\hat{\theta}))$ is the prediction error. Practical identifiability can only be assessed post estimation because it is a function of the data set, whereas sloppiness can be evaluated at any given unknown space ($\phi$), which includes parameter space and data set. Ideally, the infinite width of the confidence interval of a parameter estimate is considered practically unidentifiable. However, in practical scenarios, significant standard errors in the estimates are considered practically unidentifiable.

Consider the linear predictor given in (13). Choose another parameter set $\theta_1 = \begin{bmatrix} b_1 & b_2 & \cdots & b_n \end{bmatrix}^T$ such that $\delta = \sqrt{(a_1 - b_1)^2 + \cdots + (a_n - b_n)^2}$.

$$(y(x) - y_1(x))^2 = ((a_1 - b_1)x_1 + (a_2 - b_2)x_2 + \cdots + (a_n - b_n)x_n)^2 \tag{17}$$

$$\epsilon = (\delta_1 x_1 + \delta_2 x_2 + \cdots + \delta_n x_n)^2 \tag{18}$$

If each $x_i$ is a vector of $m$ observations, then (18) becomes,

$$\epsilon = \sum_{i=1}^{m} (\delta_1 x_{1_i} + \delta_2 x_{2_i} + \cdots + \delta_n x_{n_i})^2 \tag{19}$$

Inverting the diagonals of (4) gives the covariance matrix for the parameter estimates. The standard error of a parameter $a_i$ is given by

$$\sigma_{a_i} = \sqrt{\frac{1}{\sum_{i=1}^{m} x_i^2}} \tag{20}$$

Using (19), the relationship between the sloppiness and practical identifiability is derived as

$$\sigma_{a_i} = \frac{\delta_i}{\left(\sqrt{\epsilon - \left(\left(\sum_{k=1}^{m} \delta_k x_k\right)^2 + 2\sum_{i=1}^{m} (\delta_1 \delta_2 x_{1_i} x_{2_i} + \cdots)\right)}\right)}; k \neq i \tag{21}$$

This relationship holds if the data is derived from a Gaussian distribution and the least-squares cost function. From (21), it is clear that there is a relationship between sloppiness and practical identifiability. The corresponding parameter will become unidentifiable for a small $\epsilon$ and a very large $\delta_i$. The above result agrees with the result obtained in (17). For a generalised non-linear predictor, it is challenging to construct a relationship analytically. Hence, we propose a numerical method to analyse non-linear predictors for sloppiness and identifiability.

## Equivalence between traditional sloppiness and new definition of sloppiness

While traditional sloppiness measures the relative skewness between the largest and smallest eigenvalues for infinitesimal parameter perturbation, the proposed definition measures the skewness between the prediction and the variation in the parameter for non-infinitesimal perturbation.

**Theorem 1**. *For a positive definite Hessian of* (10) *with distinct eigenvalues and $\delta$ being the length of largest semi-axis, $\lambda_{min}(\nabla^2(C(\theta))) \to 0$, if* $\lim_{\epsilon \to 0}$ *and* $\lim_{\delta \to \alpha}$.

*Proof.* Let

$$C(\theta) = ||y(\theta^*, t) - y(\theta_1, t)||_2^2 \tag{22}$$

The Hessian of (22) equated at $\theta^*$ is given by

$$\nabla^2 C(\theta)_{ij} = \sum_n (y(\theta^*) - y(\theta)) \frac{\partial^2 y(\theta)}{\partial \theta_i \partial \theta_j} + \sum_n \frac{\partial y(\theta)}{\partial \theta_i} \frac{\partial y(\theta)}{\partial \theta_j} \tag{23}$$

where, $n$ is the number of observations. At nominal value $\theta^*$, the $\lim_{\epsilon \to 0}$, the Hessian of the cost function can be approximated to

$$\nabla^2 C(\theta)_{ij} \approx \sum_n \frac{\partial y(\theta)}{\partial \theta_i} \frac{\partial y(\theta)}{\partial \theta_j} \tag{24}$$

The length of the largest semi-axis and (9) are related by

$$\frac{1}{\sqrt{(\lambda_{min})}} = \delta \tag{25}$$

For an arbitrarily large $\alpha$,

$$\lim_{\delta \to \alpha} \Rightarrow \lambda_{min} \to \mathbf{0}. \tag{26}$$

$\alpha$ is an arbitrarily large scalar representing the length of the largest semi-axis of the hyper ellipsoid around the parameter on which the Hessian of the cost function is evaluated. $\alpha$ cannot be infinity because that results in a singular Hessian, which leads to the loss of identifiability. This concludes the proof.

As a consequence of Theorem 1

**Corollary 1**. *For a positive definite Hessian of $C(\theta)$ with distinct eigenvalues and $\delta$ being the length of largest semi axis, when* $\lim_{\epsilon \to 0}$ *and* $\lim_{\delta \to \alpha}$. *Then, sloppiness measure (S) increases.*

*Proof.* Sloppiness measure is given by

$$S = \frac{\lambda_{min}}{\lambda_{max}} \tag{27}$$

from (17)

$$\lim_{\delta \to \alpha} \Rightarrow \frac{\lambda_{min}}{\lambda_{max}} \to 0 \tag{28}$$

Smaller the $S$, sloppier the system is. This concludes the proof.

The $(\delta, \epsilon)$ measure guarantees the existence of at least one sloppy direction. However, the proposed definition is a super-set of the existing definition. Two important distinct features of the proposed definition are (i) The proposed definition does not reflect the total aspect ratio.

Rather, it defines the region of the parameter space over which model predictions are less than an arbitrarily small value ($\epsilon$) (ii) The crucial problem in the current measure of sloppiness is that the notion of small or large eigenvalues needs to be better defined, as pointed out in [16]. The prime reason is that the parameter estimates are affected by the eigenvalue magnitude and not the ratio. For an infinitesimal perturbation, the minimum eigenvalue of the FIM/Hessian of the cost is a function of $\delta$ as shown in Theorem 1. Moreover, the boundaries of the viable space of parameters will dictate the magnitude of $\delta$; by this, the user for a particular modelling exercise can assess the effect of eigenvalues on the precision of the parameter estimates.

## Sloppiness analysis of non-linear predictors

We provide three numerical examples to demonstrate the sloppiness analysis of nonlinear predictors. We propose a novel numerical method of analysing sloppiness based on the new definitions. The detailed working of the method is illustrated in the methods section. The first illustrative example demonstrates all three different scenarios (i) a sloppy region, (ii) a non-sloppy region and (iii) an unidentifiable region. The second example is one of the hallmark models used by Gutenkunst *et al.* in [5] to demonstrate sloppiness. The third example is a realistic pharmacodynmaic HIV infection model. Along with detecting identifiability and sensitivity analysis, we show that the proposed method also gives new biological insight into the effect of HIV infection on T-Cells. Two more examples are provided in the supplementary to demonstrate the working of the proposed examples in detecting complete unidentifiability and in a high-dimensional model.

**An illustrative example.**    In this example, we use a simple two-parameter state-space model to demonstrate the working of our method. Three different scenarios are considered to cover the loss of structural identifiability, sloppiness, and an ideal scenario. They are shown in Table 3. Consider the state-space model in (29).

$$\begin{bmatrix} \dot{x}_1 \\ \dot{x}_2 \end{bmatrix} = \begin{bmatrix} -\theta_1 & 0 \\ 0 & -\theta_2 \end{bmatrix} \begin{bmatrix} x_1 \\ x_2 \end{bmatrix} \tag{29}$$

$$y(t) = x_1(t) + x_2(t)$$
$$x_1(0) = x_2(0) = 1$$

Parameters are sampled inside an $n$-ball with radius $\delta$ around $\theta^*$. The model output $y(t, \theta)$ is computed for all the sampled parameters and sum-squared error $\gamma$ is computed. Further, the model-sensitive index is computed using (33).

Fig 5a and 5b show the plot for model sensitivity index and minimum deviation from reference $\theta^*$. In the case of the non-sloppy region, the curve in Fig 5b is monotonically increasing. The model's behaviour is distinguishable from the reference point ($\theta^*$) as $\delta$ increases. Additionally, the curve in the model sensitivity plot in Fig 5a deviates away from the unity value as $\delta$ increases. In the case of a sloppy region, in Fig 5b, the curve has an increasing trend, but the

**Table 3. Specifications of various scenarios.**

| Scenario | $\theta^*$ | $\delta$ | Region |
|:---:|:---:|:---:|:---:|
| 1 | $\theta_1 = 0.4\ \theta_2 = 1$ | 0.3 | non-sloppy |
| 2 | $\theta_1 = 1\ \theta_2 = 10$ | 0.3 | sloppy |
| 3 | $\theta_1 = 0.4\ \theta_2 = 0.5$ | 0.3 | unidentifiable |

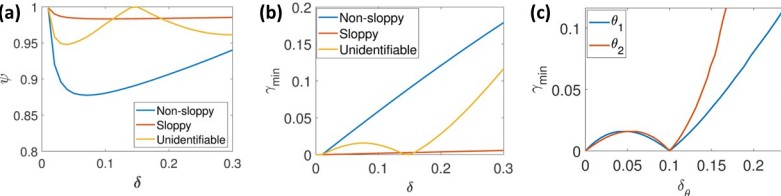

**Fig 5. Visual sensitivity analysis plot.** (a) shows the model sensitivity plot for all three regions; for the non-sloppy region, the curve deviates from the unity value for the given $\delta$, indicating local structural identifiability. The curve is very close to the unity value for the slope region, indicating an-isotropic sensitivity/ multi-scale sloppiness. The curve numerically hits unity for the unidentifiable region, indicating local structural unidentifiability for the given $\delta$. (b) shows the $\gamma_m in$ vs $\delta$ plot for all three regions; for the non-sloppy region, the $\gamma_{min}$ is numerically significant and has a constant slope for given $\delta$. For sloppy region, $\gamma_{min}$ is numerically insignificant for a significantly large $\delta$. The slope of the curve is constant, but the value of the slope is numerically closer to the zero value indicating a sloppy region in the given $\delta$. The curve numerically hits zero value respectively at $\delta = 0.14$ indicating a local structural unidentifiability. (c) Shows $\gamma_{min}$ vs $\delta_{\theta_i}$ for unidentifiable region, in this case both the $\gamma_{min}$ is zero for non-zero $\delta_{\theta_i}$ indicating local structural unidentifibility for both the parameters.

curve is almost flat and not distinguishable as compared to the non-sloppy region from the reference point. In the third scenario, where the system is locally structurally unidentifiable, from Fig 5b, it is seen that $\gamma_{min}$ is very close to zero/ numerically zero for a non-zero $\delta$. From Fig 5c, we can infer that both the parameters are unidentifiable in the region because at an absolute distance $\delta_i$ from the reference value, the prediction error goes to zero, which indicates that there is another parameter $\theta \neq \theta^*$ for which the prediction error is zero. In addition to that, in Fig 5a, the curve in model sensitivity plot touches the unity value for a non-zero $\delta$.

In order to assess local structural non-identifiability, the model sensitivity plot in conjunction with $\gamma_{min}$ plot may be used. However, a numerically zero value of $\gamma_{min}$ for a non-zero $\delta$ is sufficient to assess local structural non-identifiability. The model sensitivity index ($\psi$) quantifies the asymmetry between the most sensitive and least sensitive parameter directions in the parameter space. For a given prediction error $\epsilon = \gamma_{min}$, the conditional sloppiness can be assessed by the pair ($\epsilon, \delta$). For a given $\delta$, which can be chosen as a function of acceptable parameter range among the set of parameters, if the $\gamma_{min}$ is too low, then we can say the system is conditionally sloppy with a high probability of practically unidentifiable parameters.

**Mitotic oscillator.** The minimal cascade model for the mitotic oscillator is an ODE model with three states and ten parameters. This model is one of the sixteen system biology models that were shown to be sloppy [5]. The original model and the nominal parameters are obtained from [38]. Here, we analyze the behaviour of the model by constructing the visual plot.

The true parameters around which the model's behaviour is analyzed are taken from [38]. It can be observed from Fig 6a that for $\delta < 0.1$, there is a huge asymmetry between the

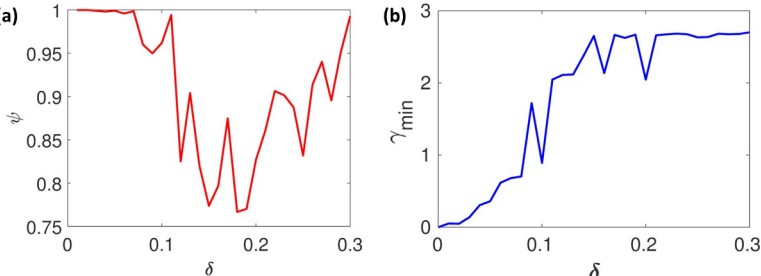

**Fig 6.** (a) The system is initially sloppy till $0 < \delta < 0.03$ and becomes non-sloppy for $0.05 < \delta < 0.15$ and again sloppy for $0.15 < \delta < 0.3$ (b) the system is locally structurally identifiable.

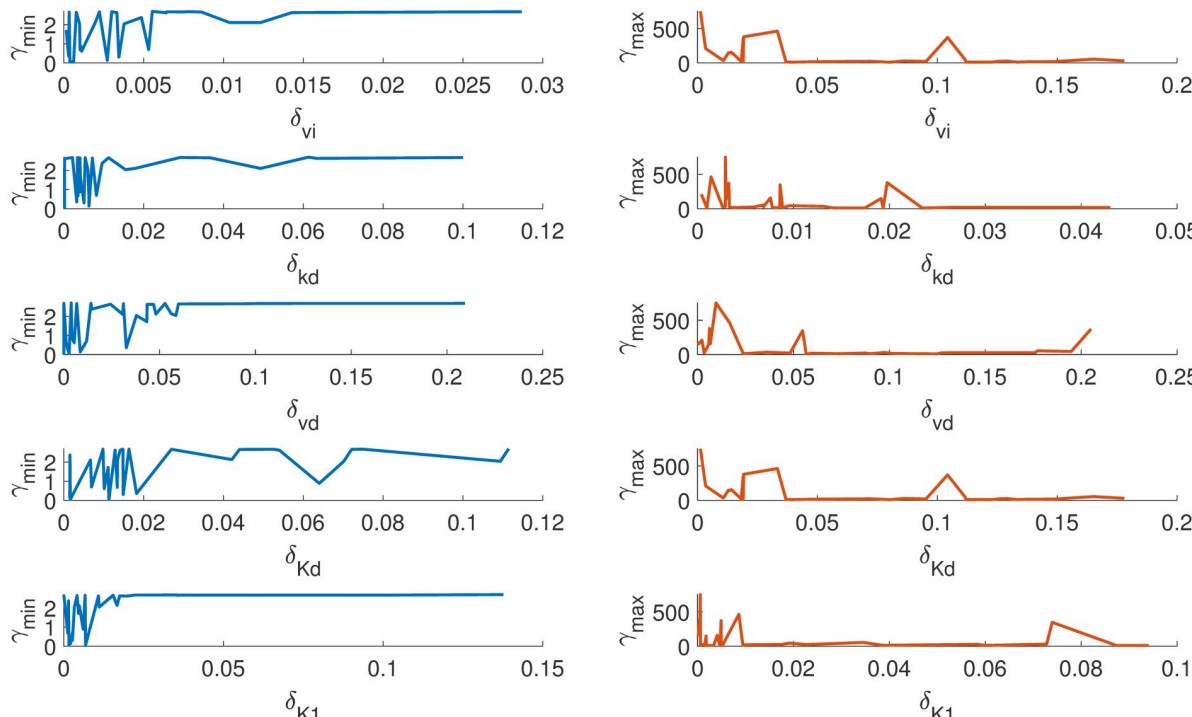

**Fig 7. $\gamma_{min}$ and $\gamma_{max}$ changes with respect to the absolute change in each parameter $v_i, k_d, v_d, K_d, K_1$ from reference values.** The parameters $K1$ and $v_d$ are highly insensitive within the given $\delta$.

minimum and maximum deviation, which implies that the system will be extremely sloppy with respect to the traditional definition of sloppiness (8) and multi-scale sloppiness [15]. From Fig 6b, it can be seen that as $\delta$ increases, $\gamma_{min}$ also increases but goes nearly flat after $\delta >$ 0.12, indicating negligible change in the $\gamma_{min}$ as $\delta$ changes, which implies that the model is ($\delta$, $\epsilon$) sloppy for $\delta > 0.12$.

Figs 7 and 8 show how $\gamma_{min}$ and $\gamma_{max}$ changes for each parameter as the absolute value changes. It can be observed the system is insensitive to parameters $k_d$, $v_d$, $K_1$, $K_2$ and $V_4$ as there is no significant change in $\gamma_{min}$ for the relative change in $\delta_i > 0.01$. On the other hand, the parameters $K_3$, $V_i$, $V_2$ and $V_4$ are highly sensitive for $\delta_i < 0.01$. This gives us a good idea of how the system behaves with respect to the changes in specific parameter intervals. This can be used to fix the initial values of the parameter during an estimation exercise in order to avoid the sloppy region, which is one of the crucial challenges in a sloppy model [24]. Our method can give a good range of initial parameter values for a nonlinear optimization algorithm.

In summary, we found that (i) the model is locally structurally identifiable, (ii) the model is sloppy in the traditional sense of sloppiness also from the proposed definition of sloppiness, (iii) our method has identified insensitive parameters and most sensitive parameters (iv) the proposed method has also identified an interval for each parameter ($\theta_i$) over which the model predictions are most and least sensitive.

**Analysis of a pharmacodynamic HIV infection model.** In this example, we analyze a published pharmacodynamic HIV infection model. The differential equation of the model and the nominal parameters are given in [39]. The model describes the effect of HIV infection on CD4+ T-cells. The model consists of four species: Uninfected T cells, latently infected T cells, actively infected T cells and free viruses. The model has four states and eight parameters. All

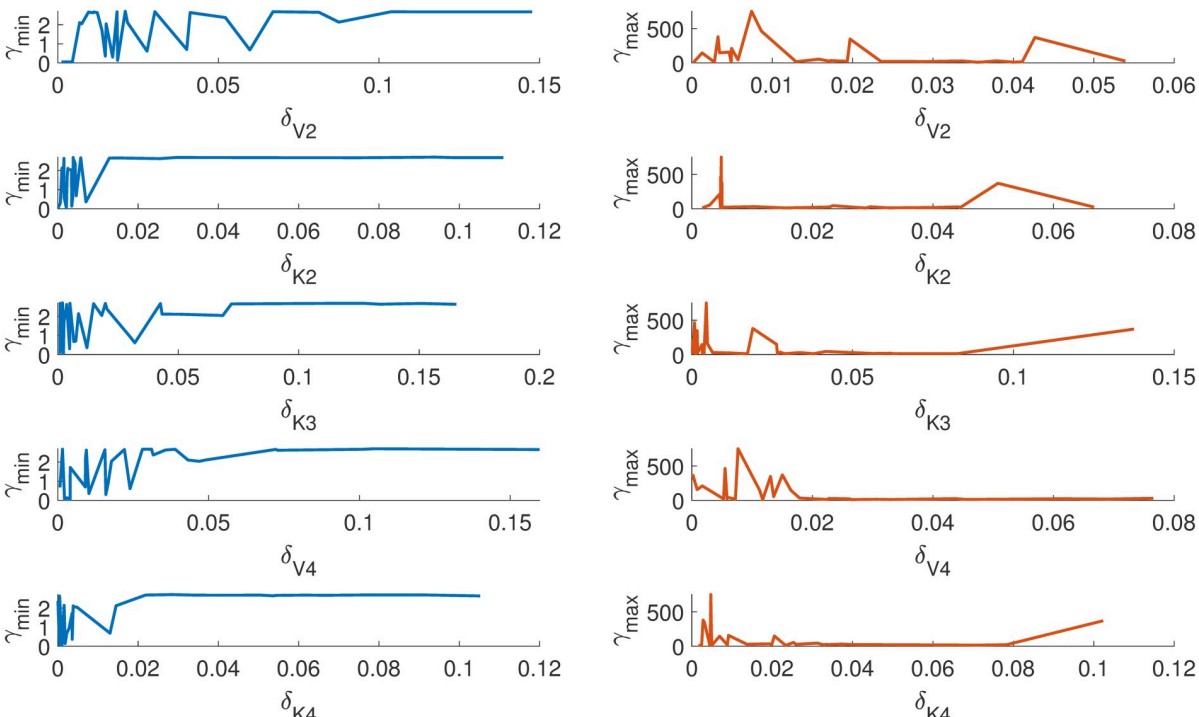

**Fig 8. The figure shows how the $\gamma_{min}$ and $\gamma_{max}$ changes with respect to the absolute change in each parameters $V_2$, $K_2$, $K_3$, $V_4$, $K_4$ from reference values.** The parameters $K_4$, $V_4$&$K_2$ are insensitive and contributes to the sloppy direction.

the parameters have biological significance. The block diagram of the model is given in Fig 9. The block diagram is adapted from [40].

In addition to detecting the local structural identifiability of the parameters, the analysis shall reveal the key parameters and critical parameter combinations that influence the free virus. The amount of free virus at any point in time influences depletion of CD4+ T cells [39] and hence finding parameters that influence free virus shall help significantly enhance the treatment regimen.

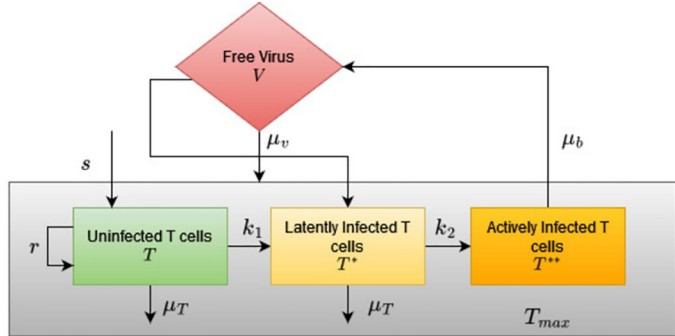

**Fig 9. The model for infection of HIV on CD4++ T-cells.** $T$- uninfected T cells, $T^*$ latently infected T cells, $T^{**}$-actively infected T-cells and $V$- free virus. The parameters include $s$—the rate of supply/production of CD4+ T-cells, $\mu_T$—the death rate of uninfected and latently infected T cells. $r$- the rate of growth of CD4+ T cell population, $k_1$-rate at which CD4+ T cells become infected, $k_2$—the rate at which latently infected T cells become actively infected, $\mu_b$-death rate of actively infected T cells, $N_v$ number of free cells produced by lysing a CD4+ T-cell, $\mu_v$—the death rate of the free virus, $T_{max}$—total number of T cells.

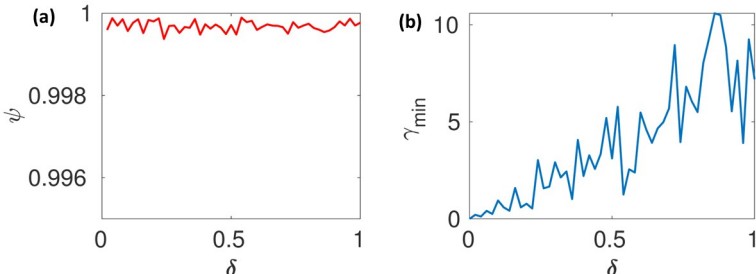

**Fig 10.** (a) The curves indicate a constant slope sensitivity ratio. (b) The curve has a significantly small slope and numerically small $\gamma_{min}$, which indicate the sloppiness for the given $\delta$ (c). The curve does not hit the unity value indicating local structural identifiability.

The parameters are transformed into log scale in order to avoid scaling issues. The initial number of samples $N_0 = 100000$ and $\alpha = 1000$. Fig 10a and 10b show the model sensitivity plot and $\gamma_{min}vs\delta$ plot. It can be observed that the model has high an isotropic sensitivity as $\psi$ is uniform and close to unity for all $\delta_i$. However $\gamma_{min}$ is significantly higher than zero for all $\delta_i$ indicating local structural identifiability inside the radius $\delta$.

Now that we know that the model has high anisotropic sensitivity inside given $\delta$, the next step is to identify the sensitive and insensitive directions and how they are distributed. Hence, we construct a heat map to characterize the directions of the most and least sensitive directions in prediction/cost space. Fig 11a shows the $cos(\theta)$ between every vector corresponding to minimum directions to every other vector and Fig 11b show the $cos(\theta)$ between every vector corresponding to maximum directions to every other vector. It is observed that the vectors corresponding to minimum deviation are spread across large dimensions. Conversely, the vectors corresponding to maximum directions are constrained to a few dimensions. This indicates that most of the directions in the parameter space are sloppy while very few contribute to the model output (free virus $V$).

Further to finding the maximum directions, we construct global sensitivity plots ($\gamma_{min}vs\theta_i$) and ($\gamma_{max}vs\theta_i$) for every parameter. From Figs 12 and 13, the parameter $\mu_b$ is the most sensitive of all the parameter followed by the parameter $r$. This indicates that a very slight movement of the maximum deviation vector in the direction of $\mu_b$(the death rate of the actively infected CD4+ T cells) creates a large change in the free virus ($V$). This result differs from the global sensitivity analysis carried out using the Sobol method in [40]. The biological implication of the result is that the death rate of actively infected T cells increases the free virus (V). This result is an interesting perspective as the death of T-cells can produce free viruses, which can

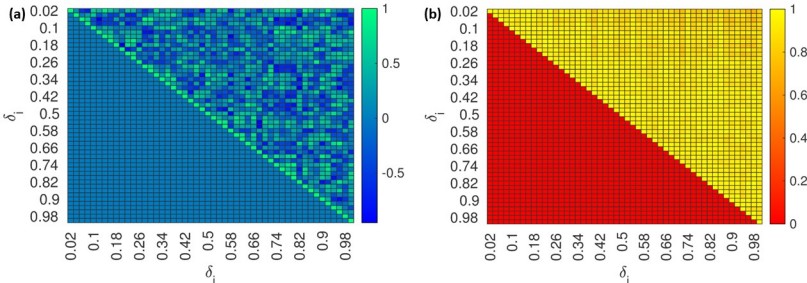

**Fig 11.** (a) The heat map indicates several minimum directions at each $\delta_i$ (b) On the other hand, the direction of maximum deviation is almost identical for each $\delta_i$.

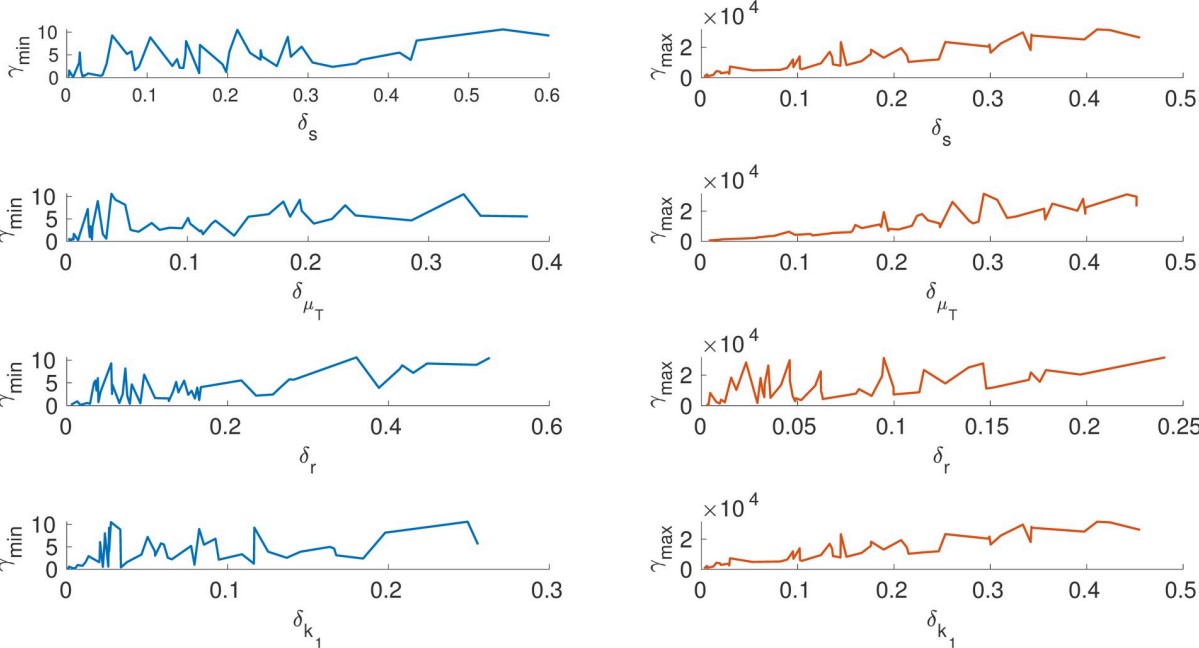

**Fig 12.** *x*-axis depicts the relative distance of the particular parameter $\theta_i$ from its reference value $\theta_i^*$. *y*-axis on the left is the minimum sum-square deviation, and on the right maximum sum-square deviation from $y^*(t)$. Parameter $r$ is more sensitive when compared to other parameters in the direction of maximum deviation.

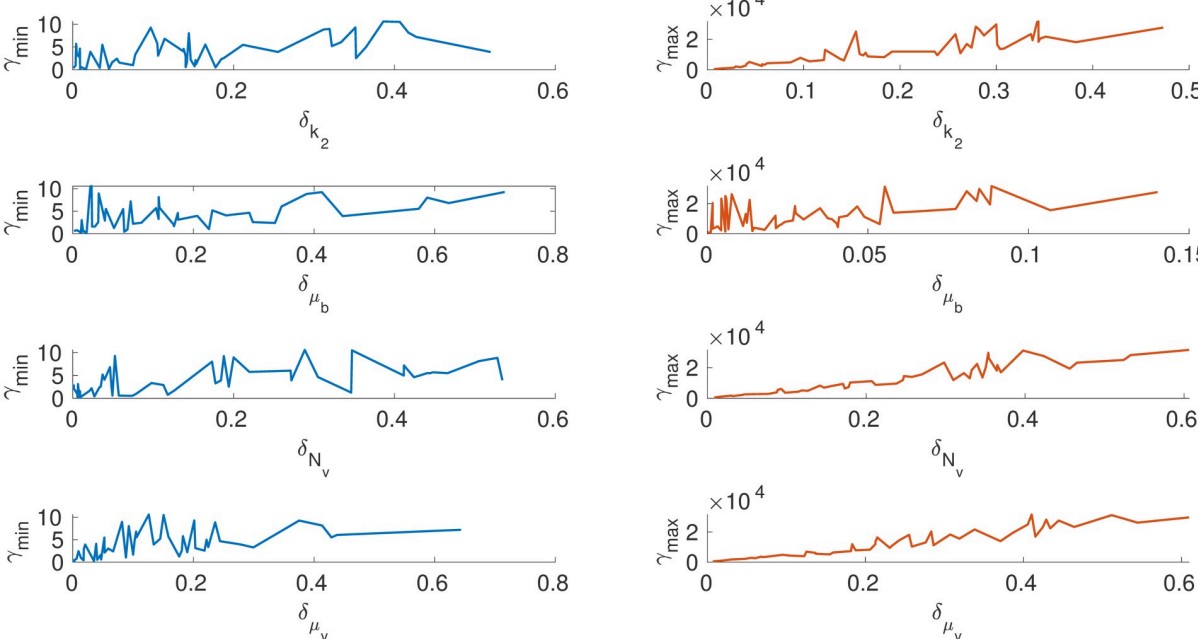

**Fig 13.** *x*-axis depicts the relative distance of the particular parameter $\theta_i$ from its reference value $\theta_i^*$. *y*-axis on the left is the minimum sum-square deviation, and on the right maximum sum-square deviation from $y^*(t)$. Parameter $\mu_b$ is sensitive compared to other parameters, and the sensitivity of the parameter $N_v$ linearly increases in the direction of maximum deviation. Summery of results is given in Table 4.

**Table 4. Summary of model features analyzed in this study.**

| Model | Parameters | Identifiable | Insensitive parameters | Sensitive parameters |
|---|---|---|---|---|
| SS: Case 1 | 2 | Yes | - | $\theta_1, \theta_2$ |
| SS: Case 2 | 2 | Yes | $\theta_2$ | $\theta_1$ |
| SS: Case 3 | 2 | No | - | - |
| Minimal Cascade | 10 | Yes | $k_d, v_d, K_1, K_2, V_4$ | $K_d, v_i, K_3$ |
| HIV Model | 8 | Yes | - | $\mu_b, r$ |

progress the diseases further. One possible suggestion from the current analysis is to delay the death rate of CD4+ T cells. From a control theory perspective, the death rate of T-cells can be used as a manipulated variable to control the free virus and disease progression in HIV infection.

## Methodology

In this section, we discuss the visual tool introduced in the previous section to detect and quantify conditional sloppiness and detect the loss of structural identifiability for generic ODE models.

### Mathematical model formulation

We consider the model of the form,

$$\dot{\mathbf{x}}(\mathbf{t}) = f(\mathbf{x}(\mathbf{t}), \theta, \mathbf{u}(\mathbf{t})) \tag{30}$$

$$\mathbf{y}(\mathbf{t}) = h(\mathbf{x}(\mathbf{t}), \theta)$$

$$\mathbf{x}(\mathbf{t_0}) = \mathbf{x_0}(\theta)$$

where $\mathbf{x} = (x_1, x_2, \ldots x_{n_x}) \in R^{n_x}$ is a state vector. $\mathbf{u} = (u_1, u_2, \ldots u_{n_u}) \in R^{n_u}$ is an $n_u$-dimensional input vector, and $\mathbf{y} = (y_1, y_2, \ldots y_{n_y}) \in R^{n_y}$ is an $n_y$-dimensional output vector. The vector $\theta = (\theta_1, \theta_2, \ldots \theta_{n_\theta}) \in R^{n_\theta}$ is the vector of parameters. The system has state function (**f**) and the observation function (**h**). The observation function can be modified by an experiment scheme, whereas the state function is fixed.

### Proposed method

The visual tool is based on the definition of sloppiness proposed in the previous section. The primary idea is to study the behaviour of the model structure around the point of interest in the parameter space. A Euclidean ball of radius $\delta$ is sampled around the parameter of interest $\theta^*$ using a multivariate Gaussian distribution. The radius $\delta$ is subdivided into $l$ equally spaced segments. The behaviour of the model is evaluated in each of the sub-Euclidean balls. The maximum and minimum deviation of predictions from the reference parameter vector is plotted against the radius vector ($\delta$).

 **Procedure.** 1. Divide the radius $\delta$ into $l$ equal segments, $\delta$ as $\delta_k$, $k = 1, 2. \ldots l$

2. Fix the sample size N for $\delta_k = 1$ and sample parameters from a Euclidean ball $B$ of radius $\delta_k$ around $\theta^*$ in the parameter space using uncorrelated multivariate Gaussian with standard

deviation as $\delta_k$.

$$B(\theta^*, \delta_k) = \{\theta | \|\theta^* - \theta\|_2 \leq \delta_k\} \tag{31}$$

3. Simulate the model output $y^*(t)$ at the optimal/true parameter $\theta^*$

4. Simulate the model output for all the parameters in the Euclidean ball and compute the sum square error $\gamma$ with the optimal output $y^*(t)$.

$$\gamma = \sum_{t=1}^{N}(y^*(t, \theta^*) - y(t, \theta))^2$$

5. Compute $\gamma_{min}$ and $\gamma_{max}$ from each increment $\delta_k$

$$\gamma_{min_k} = \min\sum_{t=1}^{N}(y^*(t, \theta^*) - y(t, \theta))^2,$$

$$\gamma_{max_k} = \max\sum_{t=1}^{N}(y^*(t, \theta^*) - y(t, \theta))^2$$

6. Update sample size

$$N(k+1) = N(k) + \alpha\left(\frac{\delta_{k+1}}{\delta_k}\right)^n \tag{32}$$

7. Repeat steps 4 to 6 while $k \leq l$.

8. Compute the model sensitivity index

$$\psi = 1 - \frac{\gamma_{min}}{\gamma_{max}} \tag{33}$$

9. Plot $(\delta, \gamma_{min})$ and $(\delta, \psi)$

Sampling $n$-ball using multivariate Gaussian distribution is one of the efficient methods. However, for a sufficiently large dimension, with a high probability, the distance between all points will be the same, and the volume of the $n$-ball goes to zero [41]. To overcome this issue, we need to generate points from independent Gaussian distribution and normalise each vector [41]. While increasing the radius $\delta_k$, the sample size $N$ has to be increased to avoid missing the regions of unidentifiability. For each increment in $\delta_k$, we derived Eq (32) to update the sample size. The parameter $\alpha$ can be used as a turning parameter to attain certain accuracy in the sampling, and $n$ is the dimension of the parameter space. Fig 14 illustrates the procedure to construct the visual tool.

## Discussion

Sloppiness, practical identifiability, and structural identifiability are the most frequently encountered challenges in computational modelling, particularly in complex dynamical systems. Assessing the model structure for sloppiness and parameter unidentifiability becomes imperative for successful parameter estimation. The concept of structural identifiability is well-established in the domain of system identification, and hence a great deal of work has been done on assessing the structural identifiability of a model structure. However, most analytical methods are limited to a specific type of non-linearity and miniature models. A few

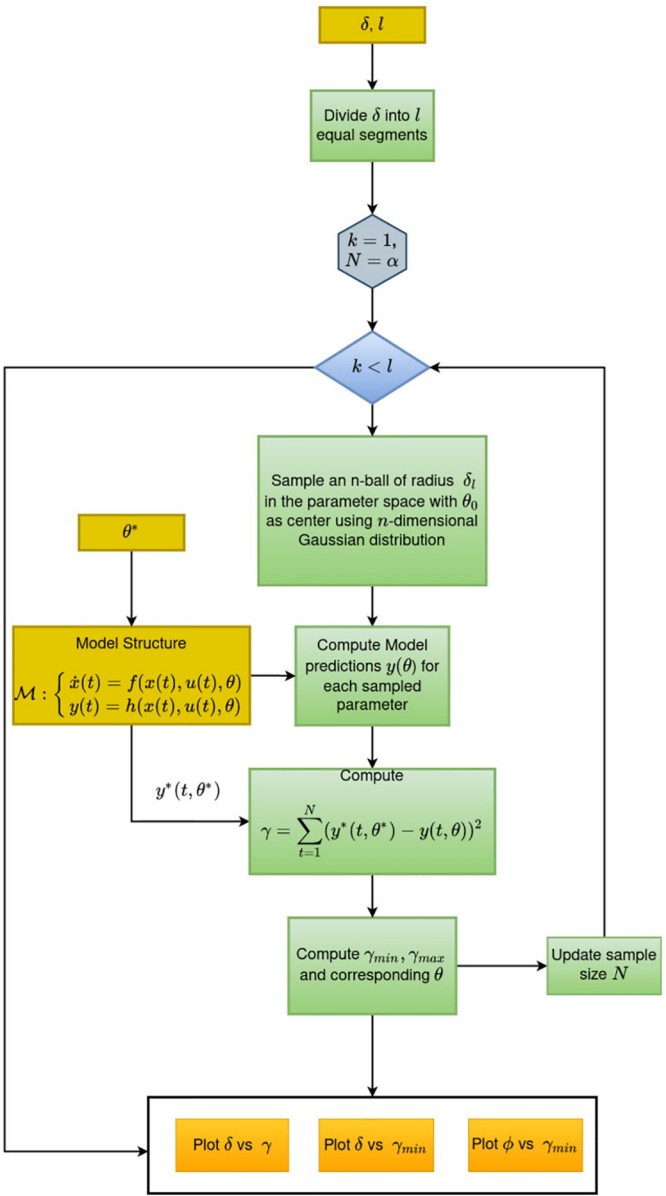

**Fig 14. Workflow to construct the visual tool.**

numerical methods also have been developed to assess structural identifiability. On the other hand, sloppiness has not been investigated with similar rigour.

Though there has been considerable discussion on sloppiness in the literature, there were a few crucial unanswered questions: (i) What is the source of sloppiness? (ii) what is the relationship of sloppiness with parameter uncertainty / practical identifiability? (iii) what is the relationship of sloppiness with inputs and estimation algorithm? This work provides definitive answers to all these questions by developing mathematical formalism of sloppiness. The new formalism defines sloppiness in an augmented space of parameters, initial conditions, and inputs.

We position the notion of sloppiness relative to well-established concepts such as structural identifiability and practical identifiability. Further, using simulations, we demonstrated the challenges in applying the current measure of sloppiness in a modelling exercise and propose

to argue that the ambiguity in understanding sloppiness and related questions is due to the lack of mathematical formalism. We developed two new theoretical definitions of sloppiness, namely sloppiness and conditional sloppiness. Conditional sloppiness is conditioned on the experiment space. Using the proposed definition of sloppiness, we showed that the linear predictors cannot be sloppy but eventually become unidentifiable. A mathematical relationship between practical identifiability and conditional sloppiness has been derived for generalized linear predictors.

A numerical method is proposed to assess the conditional sloppiness for generalized non-linear predictors. The proposed method helps determine the model's behaviour around a point of interest in the parameter space. An $n$-ball of radius $\delta$ is constructed with the reference parameter as the centre. Deviation in the prediction from the reference parameter is computed for all the parameters sampled from the $n$-ball. The minimum and maximum deviations in the prediction from the nominal parameter were plotted against the delta. The proposed tool also helps find the most sensitive and least sensitive parameters with an interval. The method can also detect structural unidentifiability.

The proposed tool was applied to three different models, including a hallmark system biology model. The proposed method detected sloppiness, structural unidentifiabilities, and sensitive and insensitive parameters. The analysis gave a holistic picture of the system's behaviour in a subset of a region in the parameters space. The parameter interval obtained from the proposed method can be used to fix the initial parameter values for the parameter optimization in sloppy models, primarily to avoid flat regions where optimization algorithms may get stuck.

In summary, we see four crucial contributions in this study (i) rightly positioning the concept of sloppiness in relationship with identifiability (ii) elucidating the challenges in the current measure of sloppiness (iii) a new mathematical definition of sloppiness, and finally, (iv) an improved framework to assess sloppiness, structural identifiability, and parameter sensitivity. We believe this work has reconciled key aspects of sloppiness in the light of system identification and will pave the way for further synergies between the systems biology and system identification communities.

## Supporting information

**S1 File. Additional examples.** This file contains the analysis of two more examples (a) High dimensional biochemical pathway to demonstrate the efficacy of the method of handle large models (b) A Cholesterol absorption model to demonstrate the ability to detect local structural unidentifiability of all the parameters.
(PDF)

**S2 File. Model equations.** The ordinary differential equation models used in the examples and the link for the code for simulating the visual plot are provided in this supporting information file.
(PDF)

## Acknowledgments

PJ acknowledges the HTRA fellowship from the Ministry of Education, Government of India. The authors thank M. Jolly and K. Hari (IISc), for useful comments on the manuscript.

## Author Contributions

**Conceptualization:** Karthik Raman, Arun K. Tangirala.

**Formal analysis:** Prem Jagadeesan.

**Funding acquisition:** Karthik Raman, Arun K. Tangirala.

**Methodology:** Prem Jagadeesan, Karthik Raman, Arun K. Tangirala.

**Supervision:** Karthik Raman, Arun K. Tangirala.

**Visualization:** Prem Jagadeesan.

**Writing – original draft:** Prem Jagadeesan.

**Writing – review & editing:** Prem Jagadeesan, Karthik Raman, Arun K. Tangirala.

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
