## [Decision Letter · Decision Letter 0]

15 Nov 2022

PONE-D-22-27480Sloppiness: fundamental study, new formalism and its application in model assessmentPLOS ONE

Dear Dr. Tangirala,

Thank you for submitting your manuscript to PLOS ONE. After careful consideration, we feel that it has merit but does not fully meet PLOS ONE’s publication criteria as it currently stands. Therefore, we invite you to submit a revised version of the manuscript that addresses the points raised during the review process.

 Both referees find the basic ideas of the paper relevant, but they have serious concerns about the major claims of the MS and also say that relevant parts of the literature were missed when writing the manuscript. Please address their main comments and answer their critical questions.

We look forward to receiving your revised manuscript.

Kind regards,

Attila Csikász-Nagy

Academic Editor

PLOS ONE

Journal Requirements:

2. Please note that PLOS ONE has specific guidelines on code sharing for submissions in which author-generated code underpins the findings in the manuscript. In these cases, all author-generated code must be made available without restrictions upon publication of the work. 

Please review our guidelines at https://journals.plos.org/plosone/s/materials-and-software-sharing#loc-sharing-code and ensure that your code is shared in a way that follows best practice and facilitates reproducibility and reuse. New software must comply with the Open Source Definition.

"NO"

"NO authors have competing interests"

6. Please ensure that you refer to Figure 12 in your text as, if accepted, production will need this reference to link the reader to the figure.

7. We note you have included a table to which you do not refer in the text of your manuscript. Please ensure that you refer to Tables 3 and 4 in your text; if accepted, production will need this reference to link the reader to the Table.

Reviewers' comments:

Reviewer's Responses to Questions

**Comments to the Author**

1. Is the manuscript technically sound, and do the data support the conclusions?

Reviewer #1: No

Reviewer #2: Partly

2. Has the statistical analysis been performed appropriately and rigorously? 

Reviewer #1: No

Reviewer #2: Yes

3. Have the authors made all data underlying the findings in their manuscript fully available?

Reviewer #1: Yes

Reviewer #2: Yes

4. Is the manuscript presented in an intelligible fashion and written in standard English?

Reviewer #1: Yes

Reviewer #2: Yes

5. Review Comments to the Author

Reviewer #1: The manuscript proposes a formal definition of sloppiness and methods to visually assess the sloppiness of a model. It then applies the method to several systems biology models.

The authors summarize their contributions as four main points, that I first address individually:

1. "rightly positioning the concept of sloppiness in relationship with identifiability."

This is indeed an important point, but the manuscript does not adequately address the previous literature on the subject. In addition, to the cited papers, the authors should consider the relation of their work to following (and papers cited therein):

Brouwer, Andrew F., and Marisa C. Eisenberg. "The underlying connections between identifiability, active subspaces, and parameter space dimension reduction." arXiv preprint arXiv:1802.05641 (2018).

Evangelou, Nikolaos, et al. "On the parameter combinations that matter and on those that do not: data-driven studies of parameter (non) identifiability." PNAS Nexus 1.4 (2022):

2. "elucidating the challenges in the current measure of sloppiness"

By this statement, I understand the authors mean to clarify that sloppiness depends on the experimental conditions and observation structure. I argue that this fact was already implicitly stated in previous measure of sloppiness. Consider, for example, the Fisher Information measures. The FIM is calculated for specific measurements at specific experimental conditions, so measures of sloppiness must also depend on these things. However, the authors are correct that this has not been widely appreciated by the community and the fact bears clarifying and repeating.

3."a new mathematical definition of sloppiness"

In my opinion, this as the most significant contribution of the paper. The definition appears to be potentially useful and applicable to model analysis as demonstrated by the authors. However, I see some shortcomings in the proposed definition that are similar to problems with other proposed sloppiness measures. Previous measures have primarily used the condition number of the FIM or the approximate model Hessian as a measure of sloppiness. However this measure only accounts for total aspect ratio (eigenvalue spread) but does not quantify whether the eigenvalues are roughly equally spaced in log. In a similar way, the proposed definition accounts for total aspect ratio but does not measure a hierarchy of aspect ratios. I would like the authors to discuss these types of limitations.

4. "a unified framework to assess sloppiness, structural identifiability, and parameter sensitivity".

This contribution is really the consequence of contributions 1 - 3 and, given my concerns stated above, is over-stated in the manuscript. I would describe the contribution as "improved methods to assess..." rather than a unified framework since the definitions proposed here are still incomplete.

More specific comments:

1. I think the Cost function Eq. (4) is incorrect. There should be off-diagonal terms coupling different x's.

2. Eq. 6 is missing signs in either the state space equations or the solution.

3. In Eq. 7, I believe the authors are reporting the Fisher Information and not the Hessian of the Cost.

4. Figures 3 -4 would be clearer on log-scales.

5. Is the ratio in the definition of S (Eq. 8) backwards. As stated, S is small for a sloppy model, but I expect it should be the other way around.

6. The new mathematical definition would be improved it were accompanied by a theorem showing a model sloppy by this definition will lead to an ill-conditioned FIM (previous definitions).

7. The literature review is missing all of the Sethna group's work from the last decade. A few papers the authors should cite and discuss include (and references therein):

Machta, Benjamin B., et al. "Parameter space compression underlies emergent theories and predictive models." Science 342.6158 (2013): 604-607.

Transtrum, Mark K., et al. "Perspective: Sloppiness and emergent theories in physics, biology, and beyond." The Journal of chemical physics 143.1 (2015): 07B201_1.

Summary:

On the whole, the paper makes a marginal contribution to the question of sloppiness/identifiability analysis in systems biology.

The literature review is inadequate and I have some concerns about the mathematical correctness of several equations and the clarity of presentation.

Reviewer #2: This is a generally well-written paper on a more formal approach of sloppiness which refers to the situation when the model output is sensitive to changes in so-called stiff parameters but largely insensitive to changes in sloppy parameters. The paper takes a further step towards formally defining sloppiness, and clarifying its relation with (practical) identifiability. The paper is well-structured and contains several useful examples illustrating the results. The results on linear models are also nice. However, there are some issues that need to be clarified before possible publication.

The parameter estimation of nonlinear dynamical models is a classical field of electrical and control engineering with clearly defined notions from well before systems biology applications. This fundamental literature is largely missing from the paper (and also from most of the papers dealing with sloppiness.) A recent survey is e.g., (Schoukens, Johan, and Lennart Ljung. "Nonlinear system identification: A user-oriented road map." IEEE Control Systems Magazine 39.6 (2019): 28-99.)

What is the benefit of sloppiness analysis in improving parameter estimation over more 'traditional' methods such as (practical) identifiability and sensitivity analysis?

Table 1 contains the weighted Hessian of $C $for assessing sloppiness. This seems to contradict Table 2 where it is indicated that $C$ does not influence sloppiness.

I cannot fully understand Eq. (9): there is $\\theta_1$ inside the norm but the $\\forall \\theta$ is used. Moreover, what is $\\mathcal{S}$?

In Definitions 1 and 2, it is not clear whether $\\epsilon$ and $\\delta$ are given when we use the defined notions. If so, why is it written that $\\epsilon$ is arbitrarily small? Or do the definitions mean that for any $\\epsilon$, there exists $\\delta$ such that the outputs will be closer than $\\epsilon$?

Please comment on why the 2-norm is used in Definitions 1 and 2. (Other norms such as the infinity norm are also frequently used for parameter estimation.)

I think the reference under Definition 2 should be Eqs. (9) and (10) instead of (7) and (9).

Why is Eq. (22) - which is an autonomous linear time invariant model - considered a non-linear predictor in the title of the section?

6. PLOS authors have the option to publish the peer review history of their article (what does this mean?). If published, this will include your full peer review and any attached files.

Reviewer #1: No

Reviewer #2: No

---

## [Author Response · Author response to Decision Letter 0]

3 Jan 2023

Reviewer 1

1. “rightly positioning the concept of sloppiness in relationship with identifiability.”

This is indeed an important point, but the manuscript does not adequately address the previous literature

on the subject. In addition, to the cited papers, the authors should consider the relation of their work to

following (and papers cited therein):

Brouwer, Andrew F., and Marisa C. Eisenberg. ”The underlying connections between identifiability, ac-

tive subspaces, and parameter space dimension reduction.” arXiv preprint arXiv:1802.05641 (2018).

Evangelou, Nikolaos, et al. ”On the parameter combinations that matter and on those that do not: data-

driven studies of parameter (non) identifiability.” PNAS Nexus 1.4 (2022):

Response: We thank the reviewer for acknowledging an important point raised in the manuscript. We also

thank the reviewer for pointing out the two references. The authors are aware of Brouwer and Eisenberg’s

work but unfortunately missed out on citing it. The revised version of the manuscript includes the relevant

references (LN:24-26 and LN:93-97)

• Brouwer and Eisenberg (2018), have used average sensitivity FIM to perform global analysis on

parameter sensitivity, sloppiness and identifiability. Further, they used the magnitude of eigenvalues

and eigenvectors of sFIM to detect the sensitivity.

• Given a nonlinear ODE, Evangelou et al (2022) developed a method to identify important and

unimportant parameter combinations using manifold learning techniques (Dmaps). Further, they

use a special type of autoencoder known as a Y-shaped conformal autoencoder to disentangle the

unimportant parameter combinations. Finally, the identified effective parameters are mapped back

to physical parameters.

2. ”elucidating the challenges in the current measure of sloppiness” By this statement, I understand the

authors mean to clarify that sloppiness depends on the experimental conditions and observation structure.

I argue that this fact was already implicitly stated in previous measure of sloppiness. Consider, for example,

the Fisher Information measures. The FIM is calculated for specific measurements at specific experimental

conditions, so measures of sloppiness must also depend on these things. However, the authors are correct

that this has not been widely appreciated by the community and the fact bears clarifying and repeating.

Response: We thank the reviewer for appreciating the need for clarity on the prevalent understanding

of the role of the model structure and data in sloppiness.

3. ”a new mathematical definition of sloppiness” In my opinion, this as the most significant contribution of

the paper. The definition appears to be potentially useful and applicable to model analysis as demonstrated

by the authors. However, I see some shortcomings in the proposed definition that are similar to problems

with other proposed sloppiness measures. Previous measures have primarily used the condition number of

the FIM or the approximate model Hessian as a measure of sloppiness. However this measure only accounts

for total aspect ratio (eigenvalue spread) but does not quantify whether the eigenvalues are roughly equally

spaced in log. In a similar way, the proposed definition accounts for total aspect ratio but does not measure

a hierarchy of aspect ratios. I would like the authors to discuss these types of limitations.

Response: Firstly, we thank the reviewer for raising this concern. We partially agree with the reviewer.

The (δ, ε) measure guarantees the existence of at least one sloppy direction. However, the proposed

definition is a super-set of the existing definition in the following two important aspects,

2

• The proposed definition does not reflect the total aspect ratio. Rather, it defines the region of the

parameter space over which model predictions are less than an arbitrarily small value (ε). The region

obtained need not always be approximated to an n-ellipsoid; it may be approximately spherical, in

which case the current measure of sloppiness might miss out on capturing it.

• The crucial problem in the traditional measure of sloppiness is that the notion of small or large

eigenvalues needs to be better defined, as pointed out by Brouwer and Eisenberg. The prime reason

is that the parameter estimates are affected by the eigenvalue magnitude and not the ratio. For an

infinitesimal perturbation, the minimum eigenvalue of the FIM/Hessian of the cost is a function of

δ as shown in Theorem 1. Moreover, the boundaries of the viable space of parameters dictate the

magnitude of δ; by this, the user for a particular modelling exercise can assess the effect of eigenvalues

on the precision of the parameter estimates.

Unfortunately, this clarity was missing in the manuscript. We regret the error. These points are now

inserted in the revised version (LN:342-353).

4. ”a unified framework to assess sloppiness, structural identifiability, and parameter sensitivity”. This

contribution is really the consequence of contributions 1 - 3 and, given my concerns stated above, is over-

stated in the manuscript. I would describe the contribution as ”improved methods to assess...” rather than

a unified framework since the definitions proposed here are still incomplete.

Response: The word unified here refers to the point that a single method detects all three aspects of

model quantification: sloppiness, identifiability and sensitivity. However, the authors also agree that the

word improved method is more appropriate than describing it as a unified method. This change has been

incorporated into the manuscript.

Specific comments

5. I think the Cost function Eq. (4) is incorrect. There should be off-diagonal terms coupling different x’s.

Response: The authors regret the mistake. The cost function has now been corrected. The impli-

cation of the corrected function on the sloppiness of the linear model has been modified accordingly in

the text. However, the inference that the Hessian of the cost function is a pure function of data still

holds. The subsequent use of Eq (4) on deriving the relationship between sloppiness and standard errors

on the parameters has not been impacted since we have considered the inversion of diagonal elements of

the Hessian as an estimate of variance (LN:146).

6. Eq. 6 is missing signs in either the state space equations or the solution.

Response: The authors regret the mistake. The negative sign is now added to the Eq. (6) (state-space

model) (LN:158).

7. In Eq. 7, I believe the authors are reporting the Fisher Information and not the Hessian of the Cost.

Response: Eq. (7) is indeed the Hessian of the cost function derived with reference to Eq. (3) in

Gutenkunst et al. Also, we agree that the Hessian of the cost function is equivalent to the Fisher In-

formation matrix when the cost function is least squares and the data falls from a Gaussian distribution.

However, since we have not considered the observation error in this model, we believe that the more

appropriate way to mention Eq. (7) is as “Hessian of the cost function” than Fisher information.

8. Figures 3 -4 would be clearer on log-scales.

Response: We thank the reviewer for the suggestion. However, in Figure 3, plotting the parameter space

3

and initial condition on a log scale makes the figure extremely skewed. For Figure 7, since the standard

error range is less than unity, the figure loses interpretability while plotting in the log scale.

9. Is the ratio in the definition of S (Eq. 8) backwards. As stated, S is small for a sloppy model, but I expect

it should be the other way around.

Response: We understand the ambiguity faced by the reviewer. In the sloppiness literature, both the

condition number and the inverse of the condition number have been used as measures. We adopted the

inverse of condition number from Christian T ¨onsing et al (2014). The prime reason for adopting the

inverse of the condition number is the interpretability in terms of identifiability. A model is known to be

unidentifiable if the Hessian of the cost/ Fisher Information Matrix (FIM) is rank deficient, which again

will result in zero value, the inverse of the condition number. Hence, the smaller the value of the inverse

of the condition number, the model is more sloppy and closer to loss of identifiability.

10. The new mathematical definition would be improved it were accompanied by a theorem showing a model

sloppy by this definition will lead to an ill-conditioned FIM (previous definitions).

Response: We thank the reviewer for this suggestion. A theorem has been added to the section

”New mathematical definition”, showing the ill-conditioning of FIM by virtue of the proposed definition

(LN:320-351).

11. The literature review is missing all of the Sethna group’s work from the last decade. A few papers the

authors should cite and discuss include (and references therein):

Machta, Benjamin B., et al. ”Parameter space compression underlies emergent theories and predictive

models.” Science 342.6158 (2013): 604-607.

Transtrum, Mark K., et al. ”Perspective: Sloppiness and emergent theories in physics, biology, and

beyond.” The Journal of Chemical Physics 143.1 (2015): 07B201 1.

Response: Authors are aware of both the articles, unfortunately we miss out on citing them. The articles

mentioned above and other relevant articles on sloppiness from the Sethna group has been cited in the

manuscript. (LN:71-72 and 83-88).

• Ben Matcha et al (2013) suggest that the sloppiness is essentially an effect of the multi-scale nature

of the underlying process. Further, they show that specific models are not sloppy at microscopic

scales, and sloppiness emerges only when fit to collective behaviour.

• Transtrum et al (2015), suggest that the existence of sloppiness (dependence of model parameters

to only a few macroscopic parameter directions) is responsible for the emergence of comprehensible

macroscopic theories from highly complex microscopic processes.

Summary:

On the whole, the paper makes a marginal contribution to the question of sloppiness/identifiability analysis

in systems biology. The literature review is inadequate and I have some concerns about the mathematical

correctness of several equations and the clarity of presentation.

Response: The mathematical correctness of the Hessian of the linear model has been addressed. The rest of

the equations are correct. With regards to the strength of contribution, the authors believe that the contribution

is significant considering the enhancements that the current work brings to the definition, overcoming of crucial

challenges in the existing measures and its potential application to model assessment.

4

Reviewer 2

1. This is a generally well-written paper on a more formal approach of sloppiness which refers to the situation

when the model output is sensitive to changes in so-called stiff parameters but largely insensitive to changes

in sloppy parameters. The paper takes a further step towards formally defining sloppiness, and clarifying

its relation with (practical) identifiability. The paper is well-structured and contains several useful examples

illustrating the results. The results on linear models are also nice. However, there are some issues that

need to be clarified before possible publication.

Response: The authors thank the reviewer for the positive feedback, and for pointing out the paper’s

key contributions. The authors agree with the issues mentioned, which have been carefully addressed in

the revised version of the manuscript.

2. The parameter estimation of nonlinear dynamical models is a classical field of electrical and control en-

gineering with clearly defined notions from well before systems biology applications. This fundamental

literature is largely missing from the paper (and also from most of the papers dealing with sloppiness.) A

recent survey is e.g., (Schoukens, Johan, and Lennart Ljung. ”Nonlinear system identification: A user-

oriented road map.” IEEE Control Systems Magazine 39.6 (2019): 28-99.)

Response: We agree with the reviewer. Challenges in parameter estimation have been systematically

studied in the domain of control engineering. The reference mentioned above has been added to the

manuscript now (LN:47-49).

3. What is the benefit of sloppiness analysis in improving parameter estimation over more ’traditional’ meth-

ods such as (practical) identifiability and sensitivity analysis?

Response: Sloppiness analysis characterizes the model’s behaviour in an entire region in the parameter

space, while identifiability and sensitivity analysis only focuses on bare parameter directions. One of the

benefits of performing sloppiness analysis is that it helps to fix the initial values of a non-linear optimiza-

tion algorithm, which directly impacts the goodness of the parameter estimates. As demonstrated in the

article “Why are nonlinear fits to data so challenging” by Transtrum et al., when the optimization gets

struck in a sloppy region, the parameter estimates thus obtained will be biased and sub-optimal. This

utility is mentioned in lines 491 to 494 in the main text.

4. Table 1 contains the weighted Hessian of Cfor assessing sloppiness. This seems to contradict Table 2

where it is indicated that C does not influence sloppiness.

Response: We thank the reviewer for pointing out an interesting observation. The authors seek to draw

the reviewer’s attention to lines 201 to 205, where we have explained why the Hessian of the cost function

has been traditionally used to assess sloppiness. A few portions have also now been rewritten for better

clarity. Sloppiness, at its core, quantifies the sensitivity of the output with respect to the parameters and

parameter directions. The influence of the cost function is indirect in that the change in the cost function

will result in a different optimal estimate. However, it will not change the geometry of the gradient of

the output with respect to the parameters/parameter directions. In Table 1, the cost function is used to

assess sloppiness because the Hessian of the cost function is representative of the gradient of the output

with respect to the parameters. The Hessian may not represent the gradient of the output when the cost

function is changed.

5. I cannot fully understand Eq. (9): there is θ1 inside the norm but the ∀θ is used. Moreover, what is S?

Response: We regret the confusion in the notation. θ1 has been changed to θ. S is a set of all parameter

vectors belong to the sloppy region. We regret missing the definition of S. It has been added now

(LN:261-269).

6. In Definitions 1 and 2, it is not clear whether ε and δ are given when we use the defined notions. If so,

why is it written that ε is arbitrarily small? Or do the definitions mean that for any ε, there exists δ such

that the outputs will be closer than ε?

Response: We would like to clarify the meaning of the definition as follows.

For a given model structure M, the experimental condition Z, for an arbitrarily small ε, if there exists

a large δ, then, the model is sloppy. The magnitude of ε and δ are user-defined and vary with the needs

of the modelling exercise. For example, ε may be chosen as the largest acceptable prediction error, and δ

can be chosen as a function of an acceptable parameter range.

7. Please comment on why the 2-norm is used in Definitions 1 and 2. (Other norms such as the infinity

norm are also frequently used for parameter estimation.)

Response: We thank the reviewer for pointing out this insight. We agree that this definition holds for

any distance measure. The reason for choosing 2-norm is that it induces a nice spherical geometry in the

parameter space, which is intuitive while assessing the model in the region of interest in the parameter

space. Moreover, while sampling high-dimensional parameter space for model assessment, approximating

the region of interest as an n-ball/n-sphere makes the sampling easier.

8. I think the reference under Definition 2 should be Eqs. (9) and (10) instead of (7) and (9).

Response: We regret the mistake in referring to equation numbers. It has been corrected (LN:270).

9. Why is Eq. (22) - which is an autonomous linear time invariant model - considered a non-linear predictor

in the title of the section?

Response: The solution to the LTI model is considered the model’s predictor equation and is nonlinear

in parameters. Hence, the solution to the autonomous LTI model is regarded as a nonlinear predictor

---

## [Decision Letter · Decision Letter 1]

25 Jan 2023

PONE-D-22-27480R1Sloppiness: fundamental study, new formalism and its application in model assessmentPLOS ONE

Dear Dr. Tangirala,

Thank you for submitting your manuscript to PLOS ONE. After careful consideration, we feel that it has merit but does not fully meet PLOS ONE’s publication criteria as it currently stands. Therefore, we invite you to submit a revised version of the manuscript that addresses the points raised during the review process.

Please answer the questions of Reviewer 2 on Theorem 1!

We look forward to receiving your revised manuscript.

Kind regards,

Attila Csikász-Nagy

Academic Editor

PLOS ONE

Journal Requirements:

Reviewers' comments:

Reviewer's Responses to Questions

**Comments to the Author**

1. If the authors have adequately addressed your comments raised in a previous round of review and you feel that this manuscript is now acceptable for publication, you may indicate that here to bypass the “Comments to the Author” section, enter your conflict of interest statement in the “Confidential to Editor” section, and submit your "Accept" recommendation.

Reviewer #1: All comments have been addressed

Reviewer #2: All comments have been addressed

2. Is the manuscript technically sound, and do the data support the conclusions?

Reviewer #1: Yes

Reviewer #2: Partly

3. Has the statistical analysis been performed appropriately and rigorously? 

Reviewer #1: Yes

Reviewer #2: Yes

4. Have the authors made all data underlying the findings in their manuscript fully available?

Reviewer #1: Yes

Reviewer #2: Yes

5. Is the manuscript presented in an intelligible fashion and written in standard English?

Reviewer #1: Yes

Reviewer #2: Yes

6. Review Comments to the Author

Reviewer #1: I thank the authors for carefully considering and incorporating my suggestions.

With the corrections to the equations and improved literature review, I believe the paper is technically correct.

It contributes an interesting perspective to the sloppiness discussion.

I recommend the paper for publication.

Reviewer #2: The manuscript has been significantly improved since the initial version, and the most important notions are more clear in the revised paper. The authors have addressed all the critical issues raised in the review. The paper gives a useful step towards understanding the relations between identifiability and sloppiness notions. The only significant problem is that the claim and the proof of Theorem 1 are not precise enough.

Major comment:

1) I cannot really follow Theorem 1 and its proof:

- What do the authors mean by "unique eigenvalues"? (Maybe distinct?).

- is $\\lambda_{min}(\\nabla (C(\\theta)))$ actually $\\lambda_{min}(\\nabla^2 (C(\\theta)))$?

- Please explain $\\alpha$.

- Eq. (23) does not seem to be correct: The Hessian should be an $n\\times n$ matrix. How can it be equal to a vector (difference of outputs)? Moreover, what is the index variable in the sums?

Please clarify these issues.

Minor comments:

2) Example 1: From a sligthly different point of view, it has been known that "it is possible for the linear predictor to show sloppiness for some experimental conditions". The condition to avoid this is called 'persistent excitation' in systems theory. See, e.g.:

(Moore, J. B. (1978). On strong consistency of least squares identification algorithms. Automatica, 14(5), 505–509. doi:10.1016/0005-1098(78)90010)

(Green, M., & Moore, J. B. (1986). Persistence of excitation in linear systems. Systems & Control Letters, 7(5), 351–360. doi:10.1016/0167-6911(86)90052)

Moreover, such sloppiness can be an important sign of over-parametrization of the linear model compared to the information content of the measurements.

3) Example 2: Initial conditions may inlfuence even the structural identifiability of dynamical models. See, e.g.:

(Saccomani, M. P., Audoly, S., & D'Angiò, L. (2003). Parameter identifiability of nonlinear systems: the role of initial conditions. Automatica, 39(4), 619-632.)

7. PLOS authors have the option to publish the peer review history of their article (what does this mean?). If published, this will include your full peer review and any attached files.

Reviewer #1: No

Reviewer #2: No

---

## [Author Response · Author response to Decision Letter 1]

1 Feb 2023

Reviewer 1

1. I thank the authors for carefully considering and incorporating my suggestions. With the corrections to

the equations and improved literature review, I believe the paper is technically correct. It contributes an

interesting perspective to the sloppiness discussion. I recommend the paper for publication.

Response: We thank the reviewer for appreciating our contribution.

Reviewer 2

1. The manuscript has been significantly improved since the initial version, and the most important notions

are more clear in the revised paper. The authors have addressed all the critical issues raised in the review.

The paper gives a useful step towards understanding the relations between identifiability and sloppiness

notions. The only significant problem is that the claim and the proof of Theorem 1 are not precise enough.

Response: The authors thank the reviewer for acknowledging the changes in the revised version of the

manuscript. The authors have addressed the reviewer’s comments below.

Major comments:

(a) What do the authors mean by ”unique eigenvalues”? (Maybe distinct?).

Response: Yes, the reviewer’s understanding is correct. The authors mean distinct eigenvalues. We

regret for the confusion caused. The word ”unique” has been changed to ”distinct” now. (Lines:329

& 341)

(b) is λmin(∇(C(θ))) actually λmin(∇2(C(θ)))?

Response: Yes, the reviewer’s understanding is correct. The authors regret the mistake. It has

been corrected in the revised version. (Line: 329)

(c) Please explain α.

Response: α is an arbitrarily large scalar representing the length of the largest semi-axis of the

hyper ellipsoid around the parameter on which the Hessian of the cost function is evaluated. α

cannot be infinity because that results in a singular Hessian, which leads to the loss of identifiability.

This explanation is now added into the manuscript (Lines: 336-338)

(d) Eq. (23) does not seem to be correct: The Hessian should be an n × n matrix. How can it be equal

to a vector (difference of outputs)? Moreover, what is the index variable in the sums? Please clarify

these issues.

Response: We regret the mistake in Eq. (23) and the subsequent derived equation Eq. (24). The

equations are corrected in the revised version. The index variable sums over the observations. (Lines:

331-332)

Theorem 1 and Corollary 1 aim to show that any model that is sloppy by Definition 1/Definition 2 will

lead to an ill-conditioned FIM. For this, we show that at an infinitesimal perturbation from the optimal/

reference parameter, the ϵ goes to zero, and the magnitude of minimum eigenvalue is δ. Further, δ

approaches some arbitrarily large value α; then the minimum goes to zero, which in turn increases the

sloppiness. Definition 1 and Definition 2 guarantee atleast one such sloppy direction.

Minor Comments

2. Example 1: From a slightly different point of view, it has been known that ”it is possible for the linear

predictor to show sloppiness for some experimental conditions”. The condition to avoid this is called

’persistent excitation’ in systems theory. See, e.g.: (Moore, J. B. (1978). On strong consistency of least

squares identification algorithms. Automatica, 14(5), 505–509. doi:10.1016/0005-1098(78)90010) (Green,

M., & Moore, J. B. (1986). Persistence of excitation in linear systems. Systems & Control Letters, 7(5),

351–360. doi:10.1016/0167-6911(86)90052) Moreover, such sloppiness can be an important sign of overparametrization

of the linear model compared to the information content of the measurements.

Response: Firstly, we thank the reviewer for the interesting insight. And we agree with the reviewer

that with persistent excitation, sloppiness can be avoided in linear least-squares problems. However, from

Eq. 4 and Eq. 5, it is seen that for any linear predictor, the source of sloppiness is data, and overparametrisation

can be seen as a consequence of fitting a higher order model when the data generating

process is lower order. We also thank the reviewer for referring to the above interesting articles.

3. Initial conditions may inlfuence even the structural identifiability of dynamical models. See, e.g.: (Saccomani,

M. P., Audoly, S., & D’Angi`o, L. (2003). Parameter identifiability of nonlinear systems: the role

of initial conditions. Automatica, 39(4), 619-632.)

Response: We agree with the reviewer; the initial conditions and input may influence structural identifiability.

The authors also cited the paper by Villaverde et al. on the role of input in structural identifiability

( “Villaverde AF, Evans ND, Chappell MJ, Banga JR. Input-Dependent Structural Identifiability of Nonlinear

Systems. IEEE Control Systems Letters. 2019;3(2):272–277.”). For autonomous systems, the initial

conditions can be seen as an impulse input at t = 0. However, while evaluating sloppiness, in this example,

we have considered the subset of parameter space and experiment space (I.C.) over which the model is

structurally identifiable. This is now mentioned in the revised manuscript (Lines 166-167), and the

reference mentioned above is now cited in the manuscript (Lines: 16-18).

---

## [Decision Letter · Decision Letter 2]

20 Feb 2023

Sloppiness: fundamental study, new formalism and its application in model assessment

PONE-D-22-27480R2

Dear Dr. Tangirala,

We’re pleased to inform you that your manuscript has been judged scientifically suitable for publication and will be formally accepted for publication once it meets all outstanding technical requirements.

Kind regards,

Attila Csikász-Nagy

Academic Editor

PLOS ONE

Additional Editor Comments (optional):

Reviewers' comments:

Reviewer's Responses to Questions

**Comments to the Author**

1. If the authors have adequately addressed your comments raised in a previous round of review and you feel that this manuscript is now acceptable for publication, you may indicate that here to bypass the “Comments to the Author” section, enter your conflict of interest statement in the “Confidential to Editor” section, and submit your "Accept" recommendation.

Reviewer #2: All comments have been addressed

2. Is the manuscript technically sound, and do the data support the conclusions?

Reviewer #2: Yes

3. Has the statistical analysis been performed appropriately and rigorously? 

Reviewer #2: Yes

4. Have the authors made all data underlying the findings in their manuscript fully available?

Reviewer #2: Yes

5. Is the manuscript presented in an intelligible fashion and written in standard English?

Reviewer #2: Yes

6. Review Comments to the Author

Reviewer #2: Theorem 1 has been clarified and therefore the paper can now be recommended for publication.

7. PLOS authors have the option to publish the peer review history of their article (what does this mean?). If published, this will include your full peer review and any attached files.

Reviewer #2: No

---

## [Editor Report · Acceptance letter]

27 Feb 2023

PONE-D-22-27480R2 

Sloppiness: fundamental study, new formalism and its application in model assessment 

Dear Dr. Tangirala:

I'm pleased to inform you that your manuscript has been deemed suitable for publication in PLOS ONE. Congratulations! Your manuscript is now with our production department. 

Kind regards, 

on behalf of

Dr. Attila Csikász-Nagy 

Academic Editor

PLOS ONE